# Scalable and Provably Fair Exposure Control
# for Large-Scale Recommender Systems

## ABSTRACT

Typical recommendation and ranking methods aim to optimize the satisfaction of users, but they are often oblivious to their impact on the items (e.g., products, jobs, news, video) and their providers. However, there has been a growing understanding that the latter is crucial to consider for a wide range of applications, since it determines the utility of those being recommended. Prior approaches to fairness-aware recommendation optimize a regularized objective to balance user satisfaction and item fairness based on some notion such as exposure fairness. These existing methods have been shown to be effective in controlling fairness, however, most of them are computationally inefficient, limiting their applications to only unrealistically small-scale situations. This indeed implies that the literature does not yet provide a solution to enable a flexible control of exposure in the industry-scale recommender systems where millions of users and items exist. To enable a computationally efficient exposure control even for such large-scale systems, this work develops a scalable, fast, and fair method called *exposure-aware ADMM (exADMM)*. Our algorithm is based on implicit alternating least squares (iALS), a conventional scalable algorithm for collaborative filtering, but optimizes a regularized objective to achieve a flexible control of accuracy-fairness tradeoff. A particular technical challenge in developing exADMM is the fact that the fairness regularizer destroys the separability of optimization subproblems for users and items, which is an essential property to ensure the scalability of iALS. Therefore, we develop a set of optimization tools to enable yet scalable fairness control with provable convergence guarantees as a basis of our algorithm. Extensive experiments performed on three recommendation datasets demonstrate that exADMM enables a far more flexible fairness control than the vanilla version of iALS, while being much more computationally efficient than existing fairness-aware recommendation methods.

## ACM Reference Format:
Anonymous Author(s). 2023. Scalable and Provably Fair Exposure Control for Large-Scale Recommender Systems. In *Proceedings of ACM Conference (Conference'17)*. ACM, New York, NY, USA, 15 pages. https://doi.org/10.1145/nnnnnnn.nnnnnnn

## 1 Introduction

Personalized recommender system has been a core function of many online platforms such as e-commerce, advertising, dating app, and online job markets. In these systems, the items to be recommended and ranked are products, job candidates, or other entities that transfer economic benefit, and it is widely recognized that how they are exposed to users has a crucial influence on their economic success [33, 44, 50]. It has also been recognized that recommender systems are responsible for and should be aware of potential societal concerns in diverse contexts, such as popularity bias [2, 3, 35, 43, 45], sales concentration in e-commerce [9, 19], filter bubbles, biased news recommendation in social media sites [23, 40], and item-side fairness in two-sided markets [1, 10]. In essence, these concerns all demand a form of *exposure control* to ensure that each item receives "fair" exposure to relevant users while not greatly sacrificing user satisfaction or recommendation accuracy. However, implementing exposure control poses technical challenges in model optimization since its objective function often becomes non-trivial and hard-to-handle in a scalable and efficient way. Building a practical system, therefore, requires a careful consideration and balance between computational requirements, user satisfaction, and item fairness via an effective and scalable exposure control.

**Related Work.** In the context of fairness-aware recommendation and ranking, there exist numerous studies on learning fair probabilistic rankings based on *pre-trained* preferences [7, 16, 17, 32, 44, 49]. The problem is often formulated as a convex optimization on doubly stochastic matrices with the size of $|\mathcal{V}| \times |\mathcal{V}|$ (where $\mathcal{V}$ is the item set) for each user. Whereas formulating ranking optimization through doubly stochastic matrices is advantageous for differentiability and convexity, this approach may not be applicable to most industry systems because of the space complexity of $O(|\mathcal{U}||\mathcal{V}|K)$ for top-$K$ recommendation (where $\mathcal{U}$ is the user set). More recent methods [16, 17, 56] are based on the Frank–Wolfe-type efficient algorithm [20, 26], which requires a top-$K$ sorting of items for each user at each iteration, resulting in a computational cost of $O(|\mathcal{U}||\mathcal{V}|\log K)$ per training epoch, which is still prohibitively high. Patro et al. [36] proposed a greedy round-robin algorithm called FairRec, which also does not scale well because its round-robin scheduling is not parallelizable. Notably, these *post-processing* methods require, a priori, a $|\mathcal{U}| \times |\mathcal{V}|$ (dense) preference matrix (e.g., a preference matrix estimated by an MF model). The preference matrix is costly to retain in the memory space or even impossible to materialize due to its cost of $O(|\mathcal{U}||\mathcal{V}|)$. Therefore, these post-processing approaches cannot exploit feedback sparsity, leading to the time and space costs of $O(|\mathcal{U}||\mathcal{V}|)$. Note that a similar approach is also adopted to control popularity bias [3, 52]. In particular, Abdollahpouri et al. [3] proposed a re-ranking algorithm based on xQuAD [47], which also suffers from a quadratic computational cost to the number of items for each user, and thus it is infeasible for large-scale systems.

In contrast to the post-processing approach, various studies have explored an *in-processing* counterpart where a single model

is trained to jointly optimize recommendation accuracy and exposure fairness [2, 11, 27, 28, 33, 34, 50, 59, 61, 62, 65]. To represent a stochastic ranking policy, several studies in information retrieval [34, 50, 61] rely on the Plackett–Luce (PL) model [37], which has a cost of $O(|\mathcal{U}||\mathcal{V}|K)$ per training epoch. When optimizing the PL model and the joint objective of ranking quality and exposure equality, stochastic gradient descent (SGD) is often applied. Although SGD allows flexible objectives and reduces the computational cost per training step, it is often difficult to apply in practice due to its severely slow convergence, particularly when the item catalogue is large [13, 63].

Compared to often inefficient fairness methods, iALS is a conventional algorithm to enable scalable recommender systems. It has also been shown by Rendle et al. [38, 39] to perform more effectively in terms of recommendation accuracy than neural collaborative filtering (NCF) [22] with a proper hyperparameter tuning. Rendle et al. [39] reported that MF models with the ALS solver [24] can be further improved by using a customized Tikhonov regularization, which is an extension of the well-known technique proposed by Zhou et al. [66]. This latest version of iALS [39] shows competitive accuracy with that of state-of-the-art methods, such as Mult-VAE [29], while substantially improving computational cost and scalability. This empirical evidence motivates us to extend iALS to build a first recommendation method that is fairness-aware and scalable to large-scale systems with over million users and items.

**Our Contributions.** To enable a scalable and provably fair exposure control for large-scale recommender systems, this work develops a new recommendation algorithm called *exposure-aware ADMM (exADMM)*, which is an extension of the celebrated iALS algorithm to achieve a scalable and flexible exposure-control. This extension is novel and non-trivial from the technical perspective, since any fairness regularizer in the objective function introduces dependency between all users and items because it involves the exposure allocation to items under the limited exposure budget. This is a major distinction from traditional personalized item recommendation since the optimal item rankings for users depend on each other when aiming for fairness. In particular, this intrinsic dependency inevitably destroys optimization separability, which is a crucial property that ensures the scalability of iALS. To overcome this technical difficulty in building a scalable method to control item fairness, we develop a set of novel optimization tools based on the alternating direction method of multipliers (ADMM) [8]. Furthermore, we provide a convergence guarantee for the proposed algorithm in terms of a non-trivial objective that includes a fairness regularizer, despite the non-convex and multi-block optimization. Finally, we provide a comprehensive empirical analysis on three datasets and demonstrate that exADMM outperforms the vanilla version of iALS in terms of fairness control while maintaining its scalability and computational efficiency. In addition, exADMM achieves similar effectiveness in terms of accuracy-fairness tradeoff compared to typical fair recommendation methods while being much more scalable and computationally faster.

Our contributions can be summarized as follows.

- We propose a first scalable method (exADMM) to enable a flexible control of accuracy-fairness tradeoff for large-scale recommender systems with over million users and items.

- We develop a set of optimization tools based on ADMM to enable an extension of iALS to an exposure-controllable variant with provable convergence guarantees.
- We empirically demonstrate that exADMM achieves similar scalability to iALS and similarly effective accuracy-fairness tradeoff compared to existing (computationally inefficient) fairness methods.

## 2 Problem Formulation and iALS

This section formulates the typical recommendation problem and the core technical details of iALS as a basis of our method.

Given users $\mathcal{U} = [|\mathcal{U}|]$ and items $\mathcal{V} = [|\mathcal{V}|]$, let $\mathbf{R} \in \{0, 1\}^{|\mathcal{U}| \times |\mathcal{V}|}$ be an implicit feedback matrix whose $(i, j)$-element has the value of 1 when user $i \in \mathcal{U}$ has interacted with item $j \in \mathcal{V}$; otherwise, it has a value of 0. We represent the number of observed interactions by that of non-zero entries in $\mathbf{R}$, which is denoted as $\text{nz}(\mathbf{R})$. iALS is an MF-based method, and its model parameters are $d$-dimensional embeddings, $\mathbf{U} \in \mathbb{R}^{|\mathcal{U}| \times d}$ and $\mathbf{V} \in \mathbb{R}^{|\mathcal{V}| \times d}$ for users and items, respectively. These parameters are typically learned by minimizing the following objective:

$$L(\mathbf{V}, \mathbf{U}) = \frac{1}{2} \left\| \mathbf{R} \odot (\mathbf{R} - \mathbf{U}\mathbf{V}^\top) \right\|_F^2 + \frac{\alpha_0}{2} \left\| \mathbf{U}\mathbf{V}^\top \right\|_F^2$$
$$+ \frac{1}{2} \left\| \mathbf{\Lambda}_U^{1/2} \mathbf{U} \right\|_F^2 + \frac{1}{2} \left\| \mathbf{\Lambda}_V^{1/2} \mathbf{V} \right\|_F^2, \tag{1}$$

where operator $\odot$ is the Hadamard element-wise product, and the second term is the L2 norm of the recovered score matrix $\mathbf{U}\mathbf{V}^\top$ (i.e., implicit regularizer [5]) with a weight parameter $\alpha_0 > 0$. $\mathbf{\Lambda}_U \in \mathbb{R}^{|\mathcal{U}| \times |\mathcal{U}|}$ and $\mathbf{\Lambda}_V \in \mathbb{R}^{|\mathcal{V}| \times |\mathcal{V}|}$ are diagonal matrices (a.k.a. Tikhonov matrices [66]) representing the weights for L2 regularization. Let $\mathbf{r}_{i,\cdot}$ and $\mathbf{r}_{\cdot,j}$ be the (column) vectors that correspond to the $i$-th row and the $j$-th column of $\mathbf{R}$, respectively. The frequency-based weighting strategy sets the weights with internal hyperparameters $\lambda_{L2} > 0$ and exponent $\eta \geq 0$ as follows:

$$(\mathbf{\Lambda}_U)_{i,i} = \lambda_{L2} \left( \left\| \mathbf{r}_{i,\cdot} \right\|_1 + \alpha_0 |\mathcal{V}| \right)^\eta, \quad (\mathbf{\Lambda}_V)_{j,j} = \lambda_{L2} \left( \left\| \mathbf{r}_{\cdot,j} \right\|_1 + \alpha_0 |\mathcal{U}| \right)^\eta.$$

Hereafter, we use $\lambda_U^{(i)} := (\mathbf{\Lambda}_U)_{i,i}$ and $\lambda_V^{(j)} := (\mathbf{\Lambda}_V)_{j,j}$.

iALS solves the minimization problem in Eq. (1) by alternating the optimization of $\mathbf{V}$ and $\mathbf{U}$. Specifically, in the $k$-th step, iALS updates $\mathbf{U}$ and $\mathbf{V}$ via

$$\mathbf{U}^{k+1} = \underset{\mathbf{U}}{\text{argmin}} \, \|\mathbf{R} \odot (\mathbf{R} - \mathbf{U}(\mathbf{V}^k)^\top)\|_F^2 + \alpha_0 \|\mathbf{U}(\mathbf{V}^k)^\top\|_F^2 + \|\mathbf{\Lambda}_U^{1/2}\mathbf{U}\|_F^2,$$

$$\mathbf{V}^{k+1} = \underset{\mathbf{V}}{\text{argmin}} \, \|\mathbf{R} \odot (\mathbf{R} - \mathbf{U}^{k+1}\mathbf{V}^\top)\|_F^2 + \alpha_0 \|\mathbf{U}^{k+1}\mathbf{V}^\top\|_F^2 + \|\mathbf{\Lambda}_V^{1/2}\mathbf{V}\|_F^2.$$

Owing to the alternating strategy, the optimization of $\mathbf{U}$ and $\mathbf{V}$ can be divided into *independent* convex problems for each row of $\mathbf{U}$ and $\mathbf{V}$. Let us use $\mathbf{u}_i \in \mathbb{R}^d$ to denote the (column) vector that corresponds to the $i$-th row of $\mathbf{U}$. Then, its update can simply be done via the following row-wise independent problem:

$$\mathbf{u}_i^{k+1} = \underset{\mathbf{u}_i}{\text{argmin}} \, \left\| \mathbf{r}_i \odot (\mathbf{r}_i - \mathbf{V}^k \mathbf{u}_i) \right\|_2^2 + \alpha_0 \left\| \mathbf{V}^k \mathbf{u}_i \right\|_2^2 + \lambda_U^{(i)} \|\mathbf{u}_i\|_2^2$$

$$= \left( \sum_{j \in \mathcal{V}} r_{i,j} \mathbf{v}_j^k (\mathbf{v}_j^k)^\top + \alpha_0 \mathbf{G}_V^k + \lambda_U^{(i)} \mathbf{I} \right)^{-1} \sum_{j \in \mathcal{V}} r_{i,j} \mathbf{v}_j^k,$$

where $\mathbf{G}_V^k = \sum_{j \in \mathcal{V}} \mathbf{v}_j^k (\mathbf{v}_j^k)^\top = (\mathbf{V}^k)^\top \mathbf{V}^k$ is the Gram matrix of the item embeddings in the $k$-th step, where $\mathbf{v}_j^k \in \mathbb{R}^d$ denotes the column vector that corresponds to the $j$-th row of $\mathbf{V}^k$. When $\mathbf{G}_V^k$ is pre-computed, the expected computational cost for each subproblem is reduced to $O((\text{nz}(\mathbf{R})/|\mathcal{V}|)d^2 + d^3)$ (a.k.a. the Gramian trick [39]), which consists of (i) computation of the Gramian for interacted items $\sum_{j \in \mathcal{V}} r_{i,j} (\mathbf{v}_j^k)(\mathbf{v}_j^k)^\top$ in $O((\text{nz}(\mathbf{R})/|\mathcal{V}|)d^2)$ and (ii) solving a linear system $\mathbf{H}^k \mathbf{u}_i^{k+1} = \sum_{i \in \mathcal{U}} r_{i,j} \mathbf{v}_j^k$, where $\mathbf{H}^k = \sum_{j \in \mathcal{V}} r_{i,j} \mathbf{v}_j^k (\mathbf{v}_j^k)^\top + \alpha_0 \mathbf{G}_V^k + \lambda_U^{(i)} \mathbf{I}$ in $O(d^3)$. Because the update of $\mathbf{V}$ is analogous to that of $\mathbf{U}$, the total cost of updating $\mathbf{U}$ and $\mathbf{V}$ is $O(\text{nz}(\mathbf{R})d^2 + (|\mathcal{U}| + |\mathcal{V}|)d^3)$. This is much lower than $O(|\mathcal{U}||\mathcal{V}|d^2 + (|\mathcal{U}| + |\mathcal{V}|)d^3)$ due to feedback sparsity, i.e., $\text{nz}(\mathbf{R}) \ll |\mathcal{U}||\mathcal{V}|$. In summary, iALS retains scalability, despite its objective involving *all* user-item pairs. The crux is that iALS exploits the Gramian trick and feedback sparsity to avoid the intractable factor $O(|\mathcal{U}||\mathcal{V}|)$.

## 3 The exADMM Algorithm

This section develops our proposed algorithm, exADMM, which is an extension of iALS to enable a scalable exposure control.

### 3.1 A Regularized Objective for Fairness

The aim of this paper is to enable a scalable control of item exposure so that individual items can receive attention from users more fairly while not sacrificing recommendation accuracy much. To this end, we consider minimizing the following regularized objective.

$$\min_{\mathbf{V}, \mathbf{U}} \underbrace{L(\mathbf{V}, \mathbf{U})}_{\text{typical prediction loss}} + \lambda_{ex} \cdot \underbrace{R_{ex}(\mathbf{V}, \mathbf{U})}_{\text{fairness regularizer}}, \qquad (2)$$

where $R_{ex}(\mathbf{V}, \mathbf{U})$ is a penalty term to induce exposure equality, and $\lambda_{ex}$ is the weight hyperparameter to control the balance between recommendation quality and exposure equality. As already discussed, the scalability of iALS is due to the simplicity of its objective. To retain this desirable property, we need to carefully define an exposure regularizer $R_{ex}(\mathbf{V}, \mathbf{U})$ in a still tractable way.

To define our regularizer, let us denote the predicted score for user $i$ and item $j$ by $\hat{r}_{i,j} = (\mathbf{U}\mathbf{V}^\top)_{i,j}$, and we predict the item ranking for user $i$ according to the decreasing order of $\{\hat{r}_{i,j} \mid j \in \mathcal{V}\}$. Evidently, there is a monotonic relationship between $\hat{r}_{i,j}$ and the amount of exposure that item $j$ will receive in a ranked list. Hence, we can evaluate exposure inequality under a recommendation model by the variability of items' scores averaged over the users, i.e., $\frac{1}{|\mathcal{U}|} \sum_{i \in \mathcal{U}} \hat{r}_{i,j}$ (this is for item $j$). There exist several possible measures of variability such as Gini indices [4, 17], standard deviation [16], and variance [60]. In this work, we consider the following second moment of the predicted scores as the regularizer $R_{ex}$.

$$R_{ex}(\mathbf{V}, \mathbf{U}) = \frac{1}{|\mathcal{V}|} \sum_{j \in \mathcal{V}} \left( \frac{1}{|\mathcal{U}|} \sum_{i \in \mathcal{U}} \hat{r}_{i,j} \right)^2 = \frac{1}{|\mathcal{V}|} \left\| \frac{1}{|\mathcal{U}|} \mathbf{V}\mathbf{U}^\top \mathbb{1} \right\|_2^2,$$

where $\mathbb{1}$ is the $|\mathcal{U}| \times 1$ column vector of which the elements are all 1. The fairness regularizer defined above is the L2 norm of the average scores predicted for the items. This is considered one of the reasonable measures of exposure inequality since it takes a large value for items whose average scores are either extremely large or small. Moreover, we can draw a clear technical distinction

between our fairness regularizer $R_{ex}(\mathbf{V}, \mathbf{U})$ and implicit regularizer $\|\mathbf{V}\mathbf{U}^\top\|_F^2$ of the vanilla iALS in Eq. (1). That is, the implicit regularizer penalizes the score $(\mathbf{U}\mathbf{V}^\top)_{i,j}$ of each user-item pair $(i,j) \in \mathcal{U} \times \mathcal{V}$ independently, whereas our (and any other) penalty term introduces a *structural dependency* into the recovered matrix $\mathbf{U}\mathbf{V}^\top$. Unfortunately, this structural dependency destroys the optimization separability with respect to the rows of $\mathbf{U}$ due to the averaged user embedding $(1/|\mathcal{U}|)\mathbf{U}^\top \mathbb{1}$ that appears in its definition. Optimizing $R_{ex}$ is thus not straightforward, particularly in large-scale settings, which motivates us to develop novel tools to handle this fairness regularizer in a scalable and provable fashion.

### 3.2 Scalable Optimization based on ADMM

To enable parallel optimization of our exposure-controllable objective in Eq. (2), we adopt an approach based on ADMM, which is an optimization framework with high parallelism [8] and has been adopted to enable scalable recommendations [14, 25, 51, 53, 54, 64]. To decouple the row- and column-wise dependencies in $\mathbf{U}$ introduced by our fairness regularizer, we first reformulate the optimization problem by introducing an auxiliary variable $\mathbf{s} \in \mathbb{R}^d$ as

$$\min_{\mathbf{V}, \mathbf{U}, \mathbf{s}} L(\mathbf{V}, \mathbf{U}) + \frac{\lambda_{ex}}{2} \|\mathbf{V}\mathbf{s}\|_2^2, \quad \text{s.t. } \mathbf{s} = \frac{1}{|\mathcal{U}|} \mathbf{U}^\top \mathbb{1}. \qquad (3)$$

Here, we replaced $(1/|\mathcal{U}|)\mathbf{U}^\top \mathbb{1}$ in the fairness regularizer with $\mathbf{s}$ while introducing an additional linear equality constraint. This can be further reformulated to the following saddle-point optimization:

$$\min_{\mathbf{V}, \mathbf{U}, \mathbf{s}} \max_{\mathbf{w}} L_\rho(\mathbf{V}, \mathbf{U}, \mathbf{s}, \mathbf{w}),$$

where

$$L_\rho(\mathbf{V}, \mathbf{U}, \mathbf{s}, \mathbf{w})$$
$$= L(\mathbf{V}, \mathbf{U}) + \frac{\lambda_{ex}}{2} \|\mathbf{V}\mathbf{s}\|_2^2 + \frac{\rho}{2} \left\| \frac{1}{|\mathcal{U}|} \mathbf{U}^\top \mathbb{1} - \mathbf{s} + \mathbf{w} \right\|_2^2 - \frac{\rho}{2} \|\mathbf{w}\|_2^2.$$

$L_\rho$ is the Lagrangian augmented by the ADMM penalty term with weight $\rho > 0$, and $\mathbf{w} \in \mathbb{R}^d$ is the dual variable (i.e., Lagrange multipliers) scaled by $1/\rho$. We can perform optimization in the $(k+1)$-th step by iteratively updating each variable as

$$\mathbf{V}^{k+1} = \underset{\mathbf{V}}{\operatorname{argmin}} L_\rho(\mathbf{V}, \mathbf{U}^k, \mathbf{s}^k, \mathbf{w}^k),$$
$$\mathbf{U}^{k+1} = \underset{\mathbf{U}}{\operatorname{argmin}} L_\rho(\mathbf{V}^{k+1}, \mathbf{U}, \mathbf{s}^k, \mathbf{w}^k),$$
$$\mathbf{s}^{k+1} = \underset{\mathbf{s}}{\operatorname{argmin}} L_\rho(\mathbf{V}^{k+1}, \mathbf{U}^{k+1}, \mathbf{s}, \mathbf{w}^k),$$
$$\mathbf{w}^{k+1} = \mathbf{w}^k + \frac{1}{|\mathcal{U}|} (\mathbf{U}^{k+1})^\top \mathbb{1} - \mathbf{s}^{k+1},$$

The update of $\mathbf{w}$ corresponds to the gradient ascent with respect to the dual problem $\max_{\mathbf{w}} \min_{\mathbf{V}, \mathbf{U}, \mathbf{s}} L_\rho(\mathbf{V}, \mathbf{U}, \mathbf{s}, \mathbf{w})$ with step size $\rho$ [8].

*3.2.1 Update of $\mathbf{V}$.* Next, we derive how to update $\mathbf{V}$ in the $(k+1)$-th step, which comprises independent optimization problems for the rows of $\mathbf{V}$. Suppose that $\mathbf{v}_j^{k+1} \in \mathbb{R}^d$ and $\mathbf{r}_{\cdot,j} \in \{0,1\}^{|\mathcal{U}|}$ are the column vectors indicating the $j$-th row of $\mathbf{V}^{k+1}$ and the $j$-th column

of $\mathbf{R}$, respectively. The update can then be performed by solving the following linear system.

$$\mathbf{v}_j^{k+1} = \underset{\mathbf{v}_j}{\operatorname{argmin}} \left\{ \frac{1}{2} \left\| \mathbf{r}_{\cdot,j} \odot (\mathbf{r}_{\cdot,j} - \mathbf{U}^k \mathbf{v}_j) \right\|_2^2 + \frac{\alpha_0}{2} \left\| \mathbf{U}^k \mathbf{v}_j \right\|_2^2 \right.$$
$$\left. + \frac{\lambda_V^{(j)}}{2} \left\| \mathbf{v}_j \right\|_2^2 + \frac{\lambda_{ex}}{2} \left( \mathbf{v}_j^\top \mathbf{s}^k \right)^2 \right\}$$
$$= \left( \sum_{i \in \mathcal{U}} r_{i,j} \mathbf{u}_i^k (\mathbf{u}_i^k)^\top + \alpha_0 \mathbf{G}_U^k + \lambda_{ex} \mathbf{s}^k (\mathbf{s}^k)^\top + \lambda_V^{(j)} \mathbf{I} \right)^{-1} \sum_{i \in \mathcal{U}} r_{i,j} \mathbf{u}_i^k,$$

where $\mathbf{G}_U^k = \sum_{i \in \mathcal{U}} \mathbf{u}_i^k (\mathbf{u}_i^k)^\top$ is the Gramian of the user embeddings in the $k$-th step. Notably, we can pre-compute $\mathbf{G}_U^k$ and $\mathbf{s}^k (\mathbf{s}^k)^\top$, and thus the update of $\mathbf{V}$ achieves the same complexity as that of iALS.

3.2.2 *Update of* $\mathbf{U}$. Updating $\mathbf{U}$ is the most intricate part of our algorithm. In the $(k+1)$-th step, our aim is to solve the following

$$\underset{\mathbf{U}}{\operatorname{argmin}} \left\{ L(\mathbf{V}^{k+1}, \mathbf{U}) + \frac{\lambda_{ex}}{2} \| \mathbf{V}^{k+1} \mathbf{s}^k \|_2^2 + \frac{\rho}{2} \left\| \frac{1}{|\mathcal{U}|} \mathbf{U}^\top \mathbb{1} - \mathbf{s}^k + \mathbf{w}^k \right\|_2^2 \right\}.$$

The issue here is that the penalty term of ADMM (the fourth term of RHS) destroys the separability regarding the rows of $\mathbf{U}$. We resolve this using a proximal gradient method [18, 30, 42]. Specifically, we consider a linear approximation (i.e., the first-order Taylor expansion around the current estimate $\mathbf{U}^k$) of the objective except for the ADMM penalization. This yields the following objective:

$$\mathbf{U}^{k+1} = \underset{\mathbf{U}}{\operatorname{argmin}} \left\{ \langle \mathbf{U} - \mathbf{U}^k, \nabla_{\mathbf{U}} g(\mathbf{V}^{k+1}, \mathbf{U}^k, \mathbf{s}^k) \rangle_F + \frac{1}{2\gamma} \left\| \mathbf{U} - \mathbf{U}^k \right\|_F^2 \right.$$
$$\left. + \frac{\rho}{2} \left\| \frac{1}{|\mathcal{U}|} \mathbf{U}^\top \mathbb{1} - \mathbf{s}^k + \mathbf{w}^k \right\|_2^2 \right\},$$

where $g(\mathbf{V}, \mathbf{U}, \mathbf{s}) = L(\mathbf{V}, \mathbf{U}) + (\lambda_{ex}/2) \| \mathbf{V} \mathbf{s} \|_2^2$. Here, we introduce a regularization term $(1/2\gamma) \| \mathbf{U} - \mathbf{U}^k \|_F^2$, which is referred to as the proximal term [42]. By completing the square, the above update can be rearranged into the following parallel and non-parallel steps:

$$\mathbf{U}^{k+1} = \underset{\mathbf{U}}{\operatorname{argmin}} \frac{\rho}{2} \left\| \frac{1}{|\mathcal{U}|} \mathbf{U}^\top \mathbb{1} - \mathbf{s}^k + \mathbf{w}^k \right\|_2^2 + \frac{1}{2\gamma} \left\| \mathbf{U} - \left( \mathbf{U}^k - \gamma \nabla_{\mathbf{U}} g^k \right) \right\|_F^2$$
$$= \underbrace{\operatorname{prox}_\gamma^k}_{\text{non-parallel}} \underbrace{(\mathbf{U}^k - \gamma \nabla_{\mathbf{U}} g^k)}_{\text{parallel}},$$

where

$$\operatorname{prox}_\gamma^k(\widetilde{\mathbf{U}}) = \underset{\mathbf{U}}{\operatorname{argmin}} \frac{\rho}{2} \left\| \frac{1}{|\mathcal{U}|} \mathbf{U}^\top \mathbb{1} - \mathbf{s}^k + \mathbf{w}^k \right\|_2^2 + \frac{1}{2\gamma} \left\| \mathbf{U} - \widetilde{\mathbf{U}} \right\|_F^2$$
$$= \left( \frac{\rho}{|\mathcal{U}|^2} \mathbb{1} \mathbb{1}^\top + \frac{1}{\gamma} \mathbf{I} \right)^{-1} \left( \frac{1}{\gamma} \widetilde{\mathbf{U}} + \frac{\rho}{|\mathcal{U}|} \mathbb{1} (\mathbf{s}^k - \mathbf{w}^k)^\top \right).$$

$\nabla_{\mathbf{U}} g^k$ is used to represent $\nabla_{\mathbf{U}} g(\mathbf{V}^{k+1}, \mathbf{U}^k, \mathbf{s}^k)$ for brevity. Note that the term $\mathbf{U}^k - \gamma \nabla_{\mathbf{U}} g^k$ corresponds to a gradient descent with respect to the iALS objective.[1] Therefore, we can update $\mathbf{U}$ in two row-wise parallel and non-parallel steps, that is, **(i)** gradient descent $\widetilde{\mathbf{U}}^{k+1} = \mathbf{U}^k - \gamma \nabla_{\mathbf{U}} g^k$ and **(ii)** proximal mapping $\mathbf{U}^{k+1} = \operatorname{prox}_\gamma^k(\widetilde{\mathbf{U}}^{k+1})$.

---

[1]Note that $\nabla_{\mathbf{U}} g(\mathbf{V}, \mathbf{U}, \mathbf{s})$ is equivalent to the gradient of $L(\mathbf{V}, \mathbf{U})$ with respect to $\mathbf{U}$ because we can ignore the constant exposure penalty $(\lambda_{ex}/2) \| \mathbf{V} \mathbf{s} \|_2^2$.

**Parallel gradient computation.** The gradient $\nabla_{\mathbf{U}} g(\mathbf{V}^{k+1}, \mathbf{U}^k, \mathbf{s}^k)$ can be independently computed for each row of $\mathbf{U}$ as follows:

$$\nabla_{\mathbf{u}_i} g^k = \left( \sum_{j \in \mathcal{V}} r_{i,j} \mathbf{v}_j^{k+1} \left( \mathbf{v}_j^{k+1} \right)^\top + \alpha_0 \mathbf{G}_V^{k+1} + \lambda_U^{(i)} \mathbf{I} \right) \mathbf{u}_i^k - \sum_{j \in \mathcal{V}} r_{i,j} \mathbf{v}_j^{k+1}.$$

Similar to iALS, we can efficiently compute the gradient by pre-computing the Gramian $\mathbf{G}_V^{k+1} = (\mathbf{V}^{k+1})^\top \mathbf{V}^{k+1}$. Thus, the gradient descent $\mathbf{U}^k - \nabla_{\mathbf{U}} g^k$ can be performed in parallel with respect to users while maintaining efficiency by exploiting the Gramian trick and feedback sparsity. It should be noted that we can avoid computing the inverse Hessian in $O(d^3)$ unlike the $\mathbf{U}$ step of iALS.

**Efficient proximal mapping.** The proximal mapping step requires a costly inversion of the $|\mathcal{U}| \times |\mathcal{U}|$ matrix in $O(|\mathcal{U}|^3)$ for a naive computation. This is problematic because, in practice, $\rho$ and $\gamma$ may increase/decrease during iterations [8]. However, we can indeed compute the inverse matrix efficiently by leveraging the Sherman-Morrison formula [48] (a special case of the Woodbury matrix identity [58]), which yields the following

$$\left( \frac{\rho}{|\mathcal{U}|^2} \mathbb{1} \mathbb{1}^\top + \frac{1}{\gamma} \mathbf{I} \right)^{-1} = - \frac{(\gamma \mathbf{I})(\rho/|\mathcal{U}|^2) \mathbb{1} \mathbb{1}^\top (\gamma \mathbf{I})}{1 + (\rho/|\mathcal{U}|^2) \mathbb{1}^\top (\gamma \mathbf{I}) \mathbb{1}} + \gamma \mathbf{I}$$
$$= \gamma \left( - \frac{1}{|\mathcal{U}|^2 \left( \frac{1}{|\mathcal{U}|} + \frac{1}{\rho \gamma} \right)} \mathbb{1} \mathbb{1}^\top + \mathbf{I} \right).$$

Therefore, we can derive the proximal mapping $\operatorname{prox}_\gamma^k$ as the following closed-form solution:

$$\operatorname{prox}_\gamma^k(\mathbf{U}) = \left( \frac{-1}{|\mathcal{U}|^2 \left( \frac{1}{|\mathcal{U}|} + \frac{1}{\rho \gamma} \right)} \mathbb{1} \mathbb{1}^\top + \mathbf{I} \right) \left( \mathbf{U} + \frac{\rho \gamma}{|\mathcal{U}|} \mathbb{1} (\mathbf{s}^k - \mathbf{w}^k)^\top \right). \tag{4}$$

The naive computation of $\operatorname{prox}_\gamma^k$ is still computationally costly due to the multiplication of $|\mathcal{U}| \times |\mathcal{U}|$ and $|\mathcal{U}| \times d$ matrices in $O(|\mathcal{U}|^2 d)$. Let us here define $\widehat{\mathbf{U}} = \mathbf{U} + (\rho \gamma/|\mathcal{U}|) \mathbb{1} (\mathbf{s}^k - \mathbf{w}^k)^\top$ for simplicity, and we can further rewrite Eq. (4) as follows:

$$\operatorname{prox}_\gamma^k(\mathbf{U}) = - \frac{1}{|\mathcal{U}|^2 \left( \frac{1}{|\mathcal{U}|} + \frac{1}{\rho \gamma} \right)} \cdot \mathbb{1} \left( \widehat{\mathbf{U}}^\top \mathbb{1} \right)^\top + \widehat{\mathbf{U}}.$$

Thus, we can perform this matrix multiplication efficiently by **(i)** computing each row of $\widehat{\mathbf{U}}$ in parallel (i.e., $\widehat{\mathbf{u}}_i = \mathbf{u}_i + (\rho \gamma/|\mathcal{U}|)(\mathbf{s}^k - \mathbf{w}^k)$), **(ii)** computing the accumulated user embedding $\mathbf{t} = \widehat{\mathbf{U}}^\top \mathbb{1}$, and then **(iii)** adding $-|\mathcal{U}|^{-2} (1/|\mathcal{U}| + 1/\rho\gamma)^{-1} \cdot \mathbf{t}$ to each row of $\widehat{\mathbf{U}}$. Thus, the cost of the matrix-matrix multiplication in Eq. (4) is reduced to $O(|\mathcal{U}|d)$, which is more efficient than $O(|\mathcal{U}|^2 d)$ of the naive implementation. The computational efficiency is advantageous even when $\rho$ and $\gamma$ are fixed during optimization.

3.2.3 *Update of* $\mathbf{s}$. We can perform the update of $\mathbf{s}$ by computing the following solution:

$$\mathbf{s}^{k+1} = \underset{\mathbf{s}}{\operatorname{argmin}} \left\{ \frac{\lambda_{ex}}{2} \left\| \mathbf{V}^{k+1} \mathbf{s} \right\|_2^2 + \frac{\rho}{2} \left\| \frac{1}{|\mathcal{U}|} (\mathbf{U}^{k+1})^\top \mathbb{1} - \mathbf{s} + \mathbf{w}^k \right\|_2^2 \right\}$$
$$= \rho \left( \lambda_{ex} \mathbf{G}_V^{k+1} + \rho \mathbf{I} \right)^{-1} \left( \frac{1}{|\mathcal{U}|} (\mathbf{U}^{k+1})^\top \mathbb{1} + \mathbf{w}^k \right).$$

**Algorithm 1** exADMM

**Require:** Implicit feedback matrix $\mathbf{R}$

1: $\forall i \in \mathcal{U}, \mathbf{u}_i^0 \sim \mathcal{N}(0, (\sigma/\sqrt{d})\mathbf{I}), \forall j \in \mathcal{V}, \mathbf{v}_j^0 \sim \mathcal{N}(0, (\sigma/\sqrt{d})\mathbf{I})$,

2: $\mathbf{s}^0 \leftarrow (1/|\mathcal{U}|)(\mathbf{U}^0)^\top \mathbb{1}, \mathbf{w}^0 \leftarrow \vec{0}$

3: **for** $k = 0, \ldots, T-1$ **do**

4: $\quad \mathbf{G}_U^k \leftarrow \sum_i \mathbf{u}_i^k (\mathbf{u}_i^k)^\top, \mathbf{G}_s^k \leftarrow \mathbf{s}^k (\mathbf{s}^k)^\top$    // $O(|\mathcal{U}|d^2)$ and $O(d^2)$

5: $\quad$ **for** $j = 1, \ldots, |\mathcal{V}|$ **do**    // ***parallelizable loop***

6: $\quad\quad \mathbf{G}_j^k \leftarrow \sum_i r_{i,j} \mathbf{u}_i^k (\mathbf{u}_i^k)^\top$    // $O((\text{nz}(\mathbf{R})/|\mathcal{V}|)d^2)$

7: $\quad\quad \mathbf{v}_j^{k+1} \leftarrow \left(\mathbf{G}_j^k + \alpha_0 \mathbf{G}_U^k + \lambda_{ex}\mathbf{G}_s^k + \lambda_V^{(j)}\mathbf{I}\right)^{-1} \sum_i r_{i,j}\mathbf{u}_i^k$    // $O(d^3)$

8: $\quad$ **end for**

9: $\quad \mathbf{G}_V^{k+1} \leftarrow \sum_j \mathbf{v}_j^{k+1}(\mathbf{v}_j^{k+1})^\top$    // $O(|\mathcal{V}|d^2)$

10: $\quad$ **for** $i = 1, \ldots, |\mathcal{U}|$ **do**    // ***parallelizable loop***

11: $\quad\quad \mathbf{G}_i^{k+1} \leftarrow \sum_j r_{i,j}\mathbf{v}_j^{k+1}(\mathbf{v}_j^{k+1})^\top$    // $O((\text{nz}(\mathbf{R})/|\mathcal{U}|)d^2)$

12: $\quad\quad \nabla_{\mathbf{u}_i} g^{k+1} \leftarrow \left(\mathbf{G}_i^{k+1} + \alpha_0 \mathbf{G}_V^{k+1} + \lambda_U^{(i)}\mathbf{I}\right)\mathbf{u}_i^k - \sum_j r_{i,j}\mathbf{v}_j^{k+1}$    // $O(d^2)$

13: $\quad\quad \widehat{\mathbf{u}}_i^{k+1} \leftarrow \mathbf{u}_i^k - \gamma \nabla_{\mathbf{u}_i} g^{k+1} + \frac{\rho\gamma}{|\mathcal{U}|}(\mathbf{s}^k - \mathbf{w}^k)$    // $O(d)$

14: $\quad$ **end for**

15: $\quad \mathbf{t} \leftarrow \sum_i \widehat{\mathbf{u}}_i^{k+1}$    // $O(|\mathcal{U}|d)$

16: $\quad$ **for** $i = 1, \ldots, |\mathcal{U}|$ **do**    // ***parallelizable loop***

17: $\quad\quad \mathbf{u}_i^{k+1} \leftarrow \widehat{\mathbf{u}}_i^{k+1} - |\mathcal{U}|^{-2}(1/|\mathcal{U}| + 1/\rho\gamma)^{-1}\mathbf{t}$    // $O(d)$

18: $\quad$ **end for**

19: $\quad \mathbf{t} \leftarrow \frac{1}{|\mathcal{U}|}\sum_i \mathbf{u}_i^{k+1}$    // $O(|\mathcal{U}|d)$

20: $\quad \mathbf{s}^{k+1} \leftarrow \rho\left(\lambda_{ex}\mathbf{G}_V^{k+1} + \rho\mathbf{I}\right)^{-1}\left(\mathbf{t} - \mathbf{w}^k\right)$    // $O(d^3)$

21: $\quad \mathbf{w}^{k+1} \leftarrow \mathbf{w}^k + \mathbf{t} - \mathbf{s}^{k+1}$    // $O(d)$

22: **end for**

23: **return** $\mathbf{U}, \mathbf{V}$

---

We can reuse the Gramian $\mathbf{G}_V^{k+1}$ for this step following its precomputation in the $\mathbf{U}$ step. The computation cost here is thus $O(|\mathcal{U}|d + d^3)$, which includes **(i)** the computation of $(1/|\mathcal{U}|)(\mathbf{U}^{k+1})^\top \mathbb{1}$ and **(ii)** the solution of a linear system of size $d^2$.

### 3.3 Complexity Analysis

Algorithm 1 shows the detailed implementation of exADMM. First, the user and item embeddings are initialized with independent normal noise with a $\sigma/\sqrt{d}$ standard deviation [39]. In line 10 of Algorithm 1, we pre-compute $\mathbf{G}_V^k = \sum_{j \in \mathcal{V}} \mathbf{v}_j^k (\mathbf{v}_j^k)^\top$, which can be done in parallel for each item and be reused in the update of $\mathbf{U}$ and $\mathbf{s}$. The averaged user vector $\mathbf{t} = (1/|\mathcal{U}|)(\mathbf{U})^\top \mathbb{1}$ can be reused for the $\mathbf{s}$ and $\mathbf{w}$ steps, hence, we compute this in line 20. Consequently, the computational costs for updating $\mathbf{V}, \mathbf{U}, \mathbf{s}$, and $\mathbf{w}$ are **(i)** $O(\text{nz}(\mathbf{R})d^2 + |\mathcal{V}|d^3)$, **(ii)** $O(\text{nz}(\mathbf{R})d^2 + |\mathcal{U}|d^2)$, **(iii)** $O(|\mathcal{U}|d + d^3)$, and **(iv)** $O(d)$, respectively. Therefore, the overall cost is $O(\text{nz}(\mathbf{R})d^2 + |\mathcal{U}|d^2 + |\mathcal{V}|d^3)$, which is indeed even lower than $O(\text{nz}(\mathbf{R})d^2 + (|\mathcal{U}| + |\mathcal{V}|)d^3)$ of iALS. This is because we avoid solving the linear system when updating $\mathbf{U}$ by applying the proximal gradient method with the efficient $\text{prox}_\gamma^k$.

### 3.4 Convergence Analysis

The objective defined in Eq. (3) has more than two variables (i.e., three-block optimization), which are *coupled* (e.g., $\mathbf{U}, \mathbf{V}$ in the iALS loss function). However, multi-block ADMM does not retain a convergence guarantee in general [12]. Various algorithms have been developed for optimization separability and provable convergence under coupled variables [15, 30, 57]. For instance, Liu et al. [30]

Table 1: Statistics of the datasets.

| Dataset | # of Users | # of Items | # of Interactions |
|---|---|---|---|
| *ML-20M* | 136,677 | 20,108 | 10M |
| *MSD* | 571,355 | 41,140 | 33.6M |
| *Epinions* | 6,287 | 3,999 | 0.13M |

proposed a variant of ADMM for non-convex problems, which completely decouples variables by introducing linear approximation when updating *all* the coupled ones, thereby enabling parallel gradient descent. By contrast, exADMM applies linearization only to the $\mathbf{U}$ step and works in an alternate way. This strategy enables second-order acceleration in the update of $\mathbf{V}$ and $\mathbf{s}$, whereas this partial linearization might impair convergence at first glance. Nonetheless, the following provides a convergence guarantee for exADMM, which is our main theoretical contribution.[2]

THEOREM 3.1. *Assume that there exist constants $C_V, C_U, C_s > 0$ such that $\|\mathbf{V}^k\|_F^2 \leq C_V, \|\mathbf{U}^k\|_F^2 \leq C_U, \|\mathbf{s}^k\|_2^2 \leq C_s$ for $\forall k \geq 0$. For $\rho \geq \max\left(\frac{24\lambda_{ex}^2 C_V C_s}{\underline{\lambda}_V}, \frac{1}{2} + \sqrt{\frac{1}{4} + 6\lambda_{ex}^2 C_V^2}\right)$ and $\gamma \leq \frac{1}{\sqrt{|\mathcal{U}|}((1+\alpha_0)C_V + \bar{\lambda}_U) + 1}$, where $\bar{\lambda}_U = \max_{i \in \mathcal{U}} \lambda_U^{(i)}$ and $\underline{\lambda}_V = \min_{j \in \mathcal{V}} \lambda_V^{(j)}$, the augmented Lagrangian $L_\rho(\mathbf{V}^k, \mathbf{U}^k, \mathbf{s}^k, \mathbf{w}^k)$ converges to some value, and residual norms $\|\mathbf{V}^{k+1} - \mathbf{V}^k\|_F, \|\mathbf{U}^{k+1} - \mathbf{U}^k\|_F, \|\mathbf{s}^{k+1} - \mathbf{s}^k\|_2$, and $\|\mathbf{w}^{k+1} - \mathbf{w}^k\|_2$ converge to 0. Furthermore, the gradients of $L_\rho$ with respect to $\mathbf{V}, \mathbf{U}, \mathbf{s}$, and $\mathbf{w}$ converge to 0.*

Theorem 3.1 illustrates that the sequence $\{\mathbf{U}^k, \mathbf{V}^k, \mathbf{s}^k, \mathbf{w}^k\}$ will converge to the feasible set, in which $\mathbf{s} = (1/|\mathcal{U}|)\mathbf{U}^\top \mathbb{1}$ in Eq. (3) holds. Moreover, the derivative of the augmented Lagrangian with respect to the primal variables (i.e., $\mathbf{V}, \mathbf{U}$, and $\mathbf{s}$) will converge to zero, which implies that the limit points of $\{\mathbf{U}^k, \mathbf{V}^k, \mathbf{s}^k, \mathbf{w}^k\}$ should be the saddle points, i.e., the KKT points of Eq. (3) if there exist. Notably, the above convergence relies on the fact that the objective is strongly convex with respect to each variable when the other variables are fixed. This property is inherited from iALS, and therefore exADMM takes advantage of iALS in both scalability and convergence guarantee while enabling flexible control of accuracy-fairness tradeoff via a (seemingly) hard-to-optimize regularizer $R_{ex}(\mathbf{V}, \mathbf{U})$.

## 4 Empirical Evaluation

This section empirically compares exADMM with iALS and existing fair recommendation methods regarding their effectiveness in accuracy-fairness control and scalability. Our experiment code is available at https://anonymous.4open.science/r/exADMM-57E4 and will be made public on Github upon publication.

### 4.1 Experiment Design

**Datasets.** Our experiments use MovieLens 20M (*ML-20M*) [21], Million Song Dataset (*MSD*) [6], and the Epinions dataset (*Epinions*) [31]. Following the standard protocol to evaluate recommendation effectiveness [29, 39], we generate implicit feedback datasets by binarizing the raw explicit feedback data by keeping interactions with ratings of four or five for *ML-20M* and *Epinions*. For *MSD*, we

---

[2]The proofs of the theorem and related lemmas are provided in Appendix A.

**Table 2: Comparison of the Scalability of Compared Methods**

| Methods | Time Complexity | Space Complexity | Is it parallelizable across users? | Is it parallelizable across items? |
|---------|-----------------|------------------|-------------------------------------|-------------------------------------|
| iALS | $O\left(\text{nz}(\mathbf{R})d^2 + (|\mathcal{U}| + |\mathcal{V}|)d^3\right)$ | $O\left((|\mathcal{U}| + |\mathcal{V}|)d\right)$ | ✔ | ✔ |
| Mult-VAE | $O\left(|\mathcal{U}||\mathcal{V}|d\right)$ | $O\left(|\mathcal{V}|d\right)$ | ✔ | ✘ |
| FairRec | $O\left(|\mathcal{U}||\mathcal{V}|(d + \log K)\right)$ | $O\left(|\mathcal{U}||\mathcal{V}|\right)$ | ✘ | ✘ |
| Multi-FR | $O\left(\text{nz}(\mathbf{R})|\mathcal{V}| + |\mathcal{U}||\mathcal{V}|d\right)$ | $O\left(|\mathcal{V}|d\right)$ | ✔ | ✘ |
| exADMM (ours) | $O\left(\text{nz}(\mathbf{R})d^2 + |\mathcal{V}|d^3\right)$ | $O\left((|\mathcal{U}| + |\mathcal{V}|)d\right)$ | ✔ | ✔ |

use all the recorded interactions as implicit feedback. Note that, for *Epinions*, we only retain users and items with more than 20 interactions following conventional studies [2, 3]. Table 1 shows the statistics of the resulting implicit feedback datasets.

Our empirical evaluation procedure follows a strong generalization setting, in which we utilize all interactions of 80% of the users for training and consider the remaining two sets of 10% of the users as holdout splits. In the validation and testing phases, each model predicts the preference scores of all items for each user on the validation and test sets.

**Compared Methods.** Since our aim is to develop a scalable method to enable accuracy-fairness control, we compare our method against an efficient but unfair method and fair but inefficient methods. Specifically, we first include the vanilla version of iALS as a scalable baseline, which does not consider item fairness. Therefore, against this baseline, our aim is to achieve better accuracy-fairness control and similar scalability. In addition to iALS, we include Mult-VAE [29], a state-of-the-art deep recommendation method, as a reference to provide the best achievable recommendation accuracy on each dataset. Besides, we consider fairness-aware recommendation methods such as the MF-based in-processing method called *Multi-FR* [59]. We also consider a post-processing method called FairRec [36] combined with the vanilla iALS algorithm to pretrain user-item preference matrix for it. We thus call this baseline iALS+FairRec. Against these fairness-aware baselines, our aim is to achieve a similarly flexible and effective accuracy-fairness control by our algorithm, which is much more scalable and computationally efficient. Table 2 summarizes the time complexity, space complexity, and parallelizablity of each method where we can see that Mult-VAE, Multi-FR, and FairRec are particularly not scalable because their complexity depends on the problematic factor of $|\mathcal{U}||\mathcal{V}|$.

Throughout the experiments, we train iALS and exADMM for $T = 50$ training epochs, Multi-VAE for $T = 200$ epochs, and Multi-FR for $T = 500$ epochs with a constant standard deviation $\sigma = 0.1$ for initialization. We tune $\{\lambda_{L2}, \alpha_0\}$ for iALS and iALS+FairRec, $\{\lambda_{L2}, \alpha_0, \lambda_{ex}, \rho, \gamma\}$ for exADMM, and $\{p, \tau, \rho_d\}$ for Multi-FR (where $p, \tau, \rho_d$ are the user patience, temperature of smooth rank functions, and dropout rate). To ensure that $\lambda_{ex}$ and $\rho$ are scale independent of $|\mathcal{U}|$, we reparametrize $\lambda_{ex}$ and $\rho$ by $\lambda_{ex} = \lambda_{ex}^* \cdot |\mathcal{U}|^2$ and $\rho = \rho^* \cdot |\mathcal{U}|^2$, respectively, and tune $\lambda_{ex}^*$ and $\rho^*$ instead of the original ones. We implement iALS, iALS+FairRec, and exADMM based on the efficient C++ implementation provided by Rendle et al. [39][3], which is multi-threaded and uses Eigen[4]. For a fair comparison, we use frequency-based re-scaling of $\Lambda_U$ and $\Lambda_V$ [39] for both iALS

---

[3]https://github.com/google-research/google-research/tree/master/ials
[4]https://eigen.tuxfamily.org

and exADMM. Note that we adopt Denoising Auto-Encoder [29] as a backbone model of Multi-FR because it can make predictions for holdout users without costly SGD iterations in the testing phase. Mult-VAE and Multi-FR are implemented with PyTorch running on a single NVIDIA P100 GPU.

**Evaluation Metrics.** We use the normalized cumulative gain (nDCG) as the measure of recommendation accuracy. To formally define this accuracy metric, let $\mathcal{V}_i \subset \mathcal{V}$ be the held-out items that user $i$ interacts with and $\pi_i(k) \in \mathcal{V}$ be the $k$-th item in the ranked list to be evaluated for $i$. Then, we can define nDCG@$K$ as

$$\text{nDCG@}K(i, \pi_i) = \frac{\text{DCG@}K(i, \pi_i)}{\text{DCG@}K(i, \pi_i^*)}, \tag{5}$$

$$\text{where} \qquad \text{DCG@}K(i, \pi_i) = \sum_{k=1}^{K} \frac{\mathbb{I}\{\pi_i(k) \in \mathcal{V}_i\}}{\log_2(k+1)},$$

and $\pi_i^*$ is an ideal ranking for user $i$. Note here that we can interpret models of user examination (i.e., item exposure) behind accuracy measures [41, 46]. Specifically, we can interpret the weight $o(i, j; \pi_i) \geq 0$ for each item $j$ in the metric as the exposure that the item receives in the ranked list for user $i$, that is, $o(i, j; \pi_i) = \mathbb{I}\{\pi_i^{-1}(j) \leq K\}/\log_2(\pi_i^{-1}(j) + 1)$ for DCG@$K$.

In addition to the accuracy metric, we use Gini@$K$ to measure the exposure inequality based on the Gini index (or Gini mean difference) [4], which is widely used to evaluate exposure inequality in related research [17, 59]. Specifically, Gini@$K$ is defined as follows:

$$\text{Gini@}K(\mathbf{o}) = \frac{1}{2\|\mathbf{o}\|_1 |\mathcal{V}|^2} \sum_{j \in \mathcal{V}} \sum_{l \in \mathcal{V}} |o_j - o_l|, \tag{6}$$

where $\mathbf{o} \in \mathbb{R}^{|\mathcal{V}|}$ is an $|\mathcal{V}|$-dimensional vector, whose $j$-th element $o_j$ indicates the total exposure given to item $j$, i.e., $o_j = \sum_i o(i, j; \pi_i)$. In our experiments, we define $o(i, j; \pi_i) = \mathbb{I}\{\pi_i^{-1}(j) \leq K\}/\log_2(\pi_i^{-1}(j) + 1)$ following the examination model of the DCG metric. Note that a lower value of Gini@$K$ indicates that recommendations are more fair towards the items, but to do so without sacrificing accuracy and scalability is particularly challenging.

## 4.2 Results and Discussion

**Accuracy-Fairness Tradeoff.** First, we evaluate and compare how well each method can control the tradeoff between recommendation accuracy (nDCG@$K$) and exposure fairness (Gini@$K$) in Figure 1 on the three datasets and three different values of $K$. In the figures, we report the Pareto frontier of exADMM, Multi-FR, and iALS+FairRec with various hyperparameter settings obtained

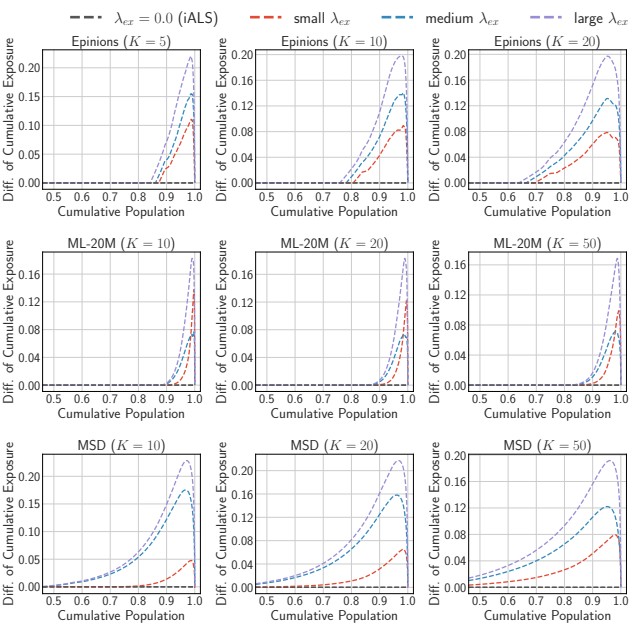

Figure 1: Tradeoff between recommendation accuracy (nDCG@K) and exposure equality (Gini@K) achieved by each method.

Figure 2: Distribution of item exposure achieved by our method with different hyperparameter ($\lambda_{ex}$) settings.

through a grid search.[5] Note that neither iALS nor Mult-VAE contains a fairness regularizer and a corresponding hyperparameter to control accuracy-fairness tradeoff in their objective, and thus we tune their hyperparameters regarding nDCG@K in the validation split and report nDCG@K and Gini@K in the test split as (blue and yellow) dots. This is why these typical methods achieve better accuracy than fairness-aware methods including exADMM, but it is also true that they produce substantial unfairness among items according to their Gini@K. It should also be noted that Multi-FR is infeasible on *ML-20M* and *MSD*, and iALS+FairRec is infeasible on *MSD*. This is due to their excessive time and space complexity, which depend on $|\mathcal{U}||\mathcal{V}|$. As a result, their results are not shown in the figures corresponding to these datasets.

From the figures, we can first see that exADMM achieves a similar accuracy-fairness tradeoff compared to Multi-FR on *Epinions* even though exADMM is much more scalable. Next, the comparisons between exADMM and iALS+FairRec (which has much larger time and space complexity compared to ours) are complicated, and it is not straightforward to determine which method performs better in terms of accuracy-fairness tradeoff. However, an interesting trend

---

[5]Hyperparameters range from $\lambda_{L2} \in [1e{-}4, 1.0]$, $\alpha_0 \in [1e{-}4, 1.0]$, $\lambda_{ex}^* \in [1e{-}10, 1e{-}4]$, $\rho^* \in [1e{-}10, 1e{-}4]$, $\gamma = \{0.001, 0.01, 0.05\}$, $\beta \in [0.1, 1.0]$, and $p \in [0.6, 1.0]$, $\tau \in [1e{-}3, 1.0]$, $\rho_d = [0.1, 0.9]$. For iALS+FairRec, we first select the best setting of $\alpha_0$ and $\lambda_{L2}$ in iALS in terms of nDCG@K and then tune the hyperparameters of FairRec, namely, the length of rankings $K$ and the scale $l \in (0, 1]$ of the minimum allocation constraint for each item $l \cdot (K \cdot |\mathcal{U}|)/|\mathcal{V}|$. We search each parameter in a logarithmic scale unless otherwise noted. For all methods, we use $d = 32$ for *Epinions*, $d = 256$ for *ML-20M*, and $d = 512$ for *MSD*.

**Table 3: Computation Time (seconds) to Complete Training and Inference of Compared Methods**

| | Epinions | | | | ML-20M | | | | MSD | | | |
| | $K = 5$ | | $K = 20$ | | $K = 10$ | | $K = 50$ | | $K = 10$ | | $K = 50$ | |
| Methods | Train | Inference | Train | Inference | Train | Inference | Train | Inference | Train | Inference | Train | Inference |
|---|---|---|---|---|---|---|---|---|---|---|---|---|
| iALS | 1.7 | 0.1 | 1.7 | 0.1 | 237.5 | 4.2 | 237.5 | 4.2 | 2,743.7 | 39.4 | 2,743.7 | 39.4 |
| Mult-VAE | 72.6 | 0.5 | 72.6 | 0.5 | 1,872.6 | 14.7 | 1,872.6 | 14.7 | 22,580.1 | 145.2 | 22,580.1 | 145.2 |
| FairRec | 1.7 | 0.3 | 1.7 | 0.3 | 237.5 | 23.3 | 237.5 | 30.1 | 878.2 | NA | 878.2 | NA |
| Multi-FR | 4,011.3 | 0.5 | 4,011.3 | 0.5 | NA | NA | NA | NA | NA | NA | NA | NA |
| exADMM (ours) | 1.7 | 0.4 | 1.7 | 0.4 | 98.0 | 9.5 | 98.0 | 9.5 | 1,533.1 | 89.6 | 1,533.1 | 89.6 |

emerges from our empirical results on *Epinions* and *ML-20M* (Fair-Rec is infeasible on *MSD*). That is, exADMM is likely to achieve a better accuracy-fairness tradeoff than iALS+FairRec when we need to achieve high recommendation accuracy, while iALS+FairRec is likely to perform better in terms of the tradeoff when we need to enforce a strong fairness requirement. This interesting difference can be attributed to the fact that exADMM is an *in-processing* method while FairRec is a *post-processing* method. Overall, it is remarkable that exADMM, which is much more scalable, achieves a competitive effectiveness in terms of accuracy-fairness control compared to Multi-FR and iALS+FairRec, which do not consider scalability.

To visualize how flexibly exADMM can control exposure distribution via its hyperparameter ($\lambda_{ex}$), Figure 2 illustrates the cumulative item exposure (called the Lorenz curve) induced by exADMM with several different values of $\lambda_{ex}$ on *Epinions* (top row), *ML-20M* (mid row), and *MSD* (bottom row). Each curve in the figures shows the cumulative item exposure relative to the case when $\lambda_{ex} = 0.0$ (i.e., the vanilla version of iALS), and this is why $\lambda_{ex} = 0.0$ (iALS) always has flat lines. In addition to the curve for $\lambda_{ex} = 0.0$, we present curves induced by three other values of $\lambda_{ex}$ (small, medium, and large) for each dataset. Specifically, we use $\lambda_{ex} = 2e{-}4$ (small), $\lambda_{ex}^* = 3e{-}3$ (medium), and $\lambda_{ex}^* = 9e{-}2$ (large) for *Epinions*, $\lambda_{ex} = 1e{-}10$, $\lambda_{ex}^* = 1e{-}7$, and $\lambda_{ex}^* = 3e{-}6$ for *ML-20M*, and $\lambda_{ex} = 1e{-}12$, $\lambda_{ex}^* = 1e{-}10$, and $\lambda_{ex}^* = 1e{-}3$ for *MSD*. Note that exADMM with the largest $\lambda_{ex}^*$ retains an acceptable recommendation accuracy for every dataset; nDCG@20 = 0.069, nDCG@50 = 0.336 on *ML-20M* and nDCG@50 = 0.236 on *MSD*, while those of iALS models are nDCG@20 = 0.080 on *Epinions*, nDCG@50 = 0.384 on *ML-20M* and nDCG@50 = 0.260 on *MSD*.

Figure 2 demonstrates that exADMM achieves fairer exposure distribution compared to iALS ($\lambda_{ex} = 0.0$). We can also see that we can flexibly and accurately control the item exposure distribution via the hyperparameter $\lambda_{ex}$ of our method. That is, we can see a monotonic relationship between $\lambda_{ex}$ and fairness of exposure distribution, suggesting that $\lambda_{ex}$ of exADMM works as an appropriate parameter to control item fairness. To sum up, Figures 1 and 2 demonstrate that exADMM enables more effective and flexible accuracy-fairness tradeoff compared to iALS and also performs competitively compared to Multi-FR and FairRec, which are computationally much more demanding. exADMM can also effectively and readily control the item exposure distribution via a scalar parameter $\lambda_{ex}$ as demonstrated in Figure 2.

**Computational Complexity.** Finally, we empirically evaluate the computational efficiency of methods in terms of both training and inference. Table 3 reports the average elapsed time to complete training and inference of each method on the three datasets and two values of $K$. Note that the "NA" values that we see for Multi-FR and FairRec indicate that their training and/or inference are infeasible.

From the table, it is evident that exADMM achieves computational efficiency equivalent to or sometimes even better than iALS across all datasets and all values of $K$, which implies that our method is able to control the accuracy-fairness tradeoff while retaining scalability.[6] We also observe that Mult-VAE needs approximately 15-40 times longer training time compared to exADMM. This issue of Mult-VAE will be exacerbated as the item space grows, given its time complexity of $O(|\mathcal{U}||\mathcal{V}|)d)$. The training procedure of Multi-FR is about 2,350 times slower than that of exADMM on *Epinions*. Besides, Multi-FR is infeasible on *ML-20M* and *MSD*. FairRec suffers from longer inference time compared to iALS and exADMM, and on *MSD*, its inference becomes infeasible. Therefore, even though Multi-FR and FairRec show their usefulness in terms of exposure control on datasets with a limited size, due to their inefficiency and large complexity (as in Table 2), they are impractical for most industry-scale systems, which can even be larger than *MSD*. This empirical observation demonstrates that exADMM is the first method that enables an effective control of accuracy-fairness tradeoff and is scalable to systems of practical size.

## 5 Conclusion

The feasibility of exposure-controllable item recommendation is indispensable for solving immediate problems of accuracy-fairness tradeoff, however, it has been disregarded in academic research. Therefore, this work studies and develops a novel scalable method, which we call exADMM, to enable flexible exposure control. Despite the technical difficulty in handling the exposure regularizer in parallel, the proposed algorithm achieves this while maintaining scalability with yet provable convergence guarantees. Empirical evaluations are promising and demonstrate that our method is the first to achieve flexible control of accuracy-fairness tradeoff in a scalable and computationally efficient way. Our work also raises several intriguing questions for future studies such as extensions to more refined fairness regularizers beyond the mere second moment, a scalable post-processing approach to control fairness, a scalable algorithm to achieve exposure fairness in two-sided preferences [55], a scalable control of impact-based fairness [44].

---

[6] exADMM can be faster than iALS due to its slightly improved computational complexity as discussed in Section 3.3 and summarized in Table 2.

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

## A Proofs of Convergence Guarantee

### A.1 Proof of 3.1

PROOF OF THEOREM 3.1. In the proof, we use the following lemma on the smoothness of $g$:

LEMMA A.1. *For any* $\mathbf{V}, \mathbf{V}' \in \mathbb{R}^{|\mathcal{V}| \times d}$, $\mathbf{s}, \mathbf{s}' \in \mathbb{R}^d$, *and* $\mathbf{U}, \mathbf{U}' \in \mathbb{R}^{|\mathcal{U}| \times d}$, *function $g$ satisfies the following inequalities:*

$$\|\nabla_{\mathbf{V}} g(\mathbf{V}, \mathbf{U}, \mathbf{s}) - \nabla_{\mathbf{V}} g(\mathbf{V}', \mathbf{U}, \mathbf{s})\|_F \leq \sqrt{|\mathcal{V}|} \left( (1 + \alpha_0) \|\mathbf{U}\|_F^2 + \lambda_{ex} \|\mathbf{s}\|_2^2 + \bar{\lambda}_V \right) \|\mathbf{V} - \mathbf{V}'\|_F,$$

$$\|\nabla_{\mathbf{U}} g(\mathbf{V}, \mathbf{U}, \mathbf{s}) - \nabla_{\mathbf{U}} g(\mathbf{V}, \mathbf{U}', \mathbf{s}')\|_F \leq \sqrt{|\mathcal{U}|} \left( (1 + \alpha_0) \|\mathbf{V}\|_F^2 + \bar{\lambda}_U \right) \|\mathbf{U} - \mathbf{U}'\|_F,$$

$$\|\nabla_{\mathbf{s}} g(\mathbf{V}, \mathbf{U}, \mathbf{s}) - \nabla_{\mathbf{s}} g(\mathbf{V}, \mathbf{U}', \mathbf{s}')\|_2 \leq \lambda_{ex} \|\mathbf{V}\|_F^2 \|\mathbf{s} - \mathbf{s}'\|_2,$$

$$\|\nabla_{\mathbf{s}} g(\mathbf{V}, \mathbf{U}, \mathbf{s}) - \nabla_{\mathbf{s}} g(\mathbf{V}', \mathbf{U}, \mathbf{s})\|_2 \leq \lambda_{ex} (\|\mathbf{V}\|_F + \|\mathbf{V}'\|_F) \|\mathbf{s}\|_2 \|\mathbf{V} - \mathbf{V}'\|_F,$$

*where* $\bar{\lambda}_U = \max_{i \in \mathcal{U}} \lambda_U^{(i)}$ *and* $\bar{\lambda}_V = \max_{j \in \mathcal{V}} \lambda_V^{(j)}$.

We prove the first part of the theorem. We decompose the difference of $L_\rho$ before and after a single epoch update into that before and after each alternating step.

$$L_\rho(\mathbf{V}^{k+1}, \mathbf{U}^{k+1}, \mathbf{s}^{k+1}, \mathbf{w}^{k+1}) - L_\rho(\mathbf{V}^k, \mathbf{U}^k, \mathbf{s}^k, \mathbf{w}^k) = \left( L_\rho(\mathbf{V}^{k+1}, \mathbf{U}^{k+1}, \mathbf{s}^k, \mathbf{w}^k) - L_\rho(\mathbf{V}^k, \mathbf{U}^k, \mathbf{s}^k, \mathbf{w}^k) \right)$$

$$+ \left( L_\rho(\mathbf{V}^{k+1}, \mathbf{U}^{k+1}, \mathbf{s}^{k+1}, \mathbf{w}^k) - L_\rho(\mathbf{V}^{k+1}, \mathbf{U}^{k+1}, \mathbf{s}^k, \mathbf{w}^k) \right)$$

$$+ \left( L_\rho(\mathbf{V}^{k+1}, \mathbf{U}^{k+1}, \mathbf{s}^{k+1}, \mathbf{w}^{k+1}) - L_\rho(\mathbf{V}^{k+1}, \mathbf{U}^{k+1}, \mathbf{s}^{k+1}, \mathbf{w}^k) \right). \tag{7}$$

Lemma A.1 implies the upper bound on each term in the RHS:

LEMMA A.2. *The update of* $\mathbf{V}$ *and* $\mathbf{U}$ *in the $(k+1)$-step satisfies*

$$L_\rho(\mathbf{V}^{k+1}, \mathbf{U}^{k+1}, \mathbf{s}^k, \mathbf{w}^k) - L_\rho(\mathbf{V}^k, \mathbf{U}^k, \mathbf{s}^k, \mathbf{w}^k) \leq \frac{\sqrt{|\mathcal{U}|}((1 + \alpha_0)C_V + \bar{\lambda}_U) - 1/\gamma}{2} \|\mathbf{U}^{k+1} - \mathbf{U}^k\|_F^2 - \frac{\lambda_V}{2} \|\mathbf{V}^{k+1} - \mathbf{V}^k\|_F^2.$$

LEMMA A.3. *The update of* $\mathbf{s}$ *in the $(k+1)$-th step satisfies*

$$L_\rho(\mathbf{V}^{k+1}, \mathbf{U}^{k+1}, \mathbf{s}^{k+1}, \mathbf{w}^k) - L_\rho(\mathbf{V}^{k+1}, \mathbf{U}^{k+1}, \mathbf{s}^k, \mathbf{w}^k) \leq -\frac{\rho}{2} \|\mathbf{s}^{k+1} - \mathbf{s}^k\|_2^2.$$

LEMMA A.4. *The update of* $\mathbf{w}$ *in the $(k+1)$-th step satisfies*

$$L_\rho(\mathbf{V}^{k+1}, \mathbf{U}^{k+1}, \mathbf{s}^{k+1}, \mathbf{w}^{k+1}) - L_\rho(\mathbf{V}^{k+1}, \mathbf{U}^{k+1}, \mathbf{s}^{k+1}, \mathbf{w}^k) \leq \frac{3\lambda_{ex}^2 C_V^2}{\rho} \|\mathbf{s}^{k+1} - \mathbf{s}^k\|_2^2 + \frac{6\lambda_{ex}^2 C_V C_s}{\rho} \|\mathbf{V}^{k+1} - \mathbf{V}^k\|_F^2.$$

By using Eq. (7) and Lemmas A.2 to A.4, under the assumptions $\rho \geq \max \left( \frac{24\lambda_{ex}^2 C_V C_s}{\lambda_V}, \frac{1}{2} + \sqrt{\frac{1}{4} + 6\lambda_{ex}^2 C_V^2} \right)$ and $\gamma \leq \frac{1}{\sqrt{|\mathcal{U}|}((1+\alpha_0)C_V + \bar{\lambda}_U) + 1}$, we have

$$L_\rho(\mathbf{V}^{k+1}, \mathbf{U}^{k+1}, \mathbf{s}^{k+1}, \mathbf{w}^{k+1}) - L_\rho(\mathbf{V}^k, \mathbf{U}^k, \mathbf{s}^k, \mathbf{w}^k)$$

$$\leq \frac{\sqrt{|\mathcal{U}|}((1 + \alpha_0)C_V + \bar{\lambda}_U) - 1/\gamma}{2} \|\mathbf{U}^{k+1} - \mathbf{U}^k\|_F^2 - \frac{\lambda_V}{2} \|\mathbf{V}^{k+1} - \mathbf{V}^k\|_F^2 - \frac{\rho}{2} \|\mathbf{s}^{k+1} - \mathbf{s}^k\|_2^2$$

$$+ \frac{3\lambda_{ex}^2 C_V^2}{\rho} \|\mathbf{s}^{k+1} - \mathbf{s}^k\|_2^2 + \frac{6\lambda_{ex}^2 C_V C_s}{\rho} \|\mathbf{V}^{k+1} - \mathbf{V}^k\|_F^2$$

$$= \frac{\sqrt{|\mathcal{U}|}((1 + \alpha_0)C_V + \bar{\lambda}_U) - 1/\gamma}{2} \|\mathbf{U}^{k+1} - \mathbf{U}^k\|_F^2 + \left( -\frac{\lambda_V}{2} + \frac{6\lambda_{ex}^2 C_V C_s}{\rho} \right) \|\mathbf{V}^{k+1} - \mathbf{V}^k\|_F^2 + \left( -\frac{\rho}{2} + \frac{3\lambda_{ex}^2 C_V^2}{\rho} \right) \|\mathbf{s}^{k+1} - \mathbf{s}^k\|_2^2$$

$$\leq -\frac{1}{2} \|\mathbf{U}^{k+1} - \mathbf{U}^k\|_F^2 - \frac{\lambda_V}{4} \|\mathbf{V}^{k+1} - \mathbf{V}^k\|_F^2 - \frac{1}{2} \|\mathbf{s}^{k+1} - \mathbf{s}^k\|_2^2 \leq 0. \tag{8}$$

Therefore, $L_\rho(\mathbf{V}^k, \mathbf{U}^k, \mathbf{s}^k, \mathbf{w}^k)$ is monotonically decreasing.

Here, we obtain the following lower bound on $L_\rho(\mathbf{V}^k, \mathbf{U}^k, \mathbf{s}^k, \mathbf{w}^k)$:

LEMMA A.5. $\mathbf{V}^k, \mathbf{U}^k, \mathbf{s}^k$, *and* $\mathbf{w}^k$ *updated by xADMM satisfy*

$$L_\rho(\mathbf{V}^k, \mathbf{U}^k, \mathbf{s}^k, \mathbf{w}^k) \geq \frac{\rho - \lambda_{ex} C_V}{2} \left\| \frac{1}{|\mathcal{U}|} (\mathbf{U}^k)^\top \mathbb{1} - \mathbf{s}^k \right\|_2^2.$$

Thus, when $\rho \geq \lambda_{ex} C_V$ holds, $L_\rho(\mathbf{V}^k, \mathbf{U}^k, \mathbf{s}^k, \mathbf{w}^k)$ is lower bounded by 0. Therefore, owing to its monotonic decrease, $L_\rho(\mathbf{V}^k, \mathbf{U}^k, \mathbf{s}^k, \mathbf{w}^k)$ converges to some constant value, and $L_\rho(\mathbf{V}^{k+1}, \mathbf{U}^{k+1}, \mathbf{s}^{k+1}, \mathbf{w}^{k+1}) - L_\rho(\mathbf{V}^k, \mathbf{U}^k, \mathbf{s}^k, \mathbf{w}^k)$ converges to 0. From Eq. (8) and the fact that $L_\rho(\mathbf{V}^{k+1}, \mathbf{U}^{k+1}, \mathbf{s}^{k+1}, \mathbf{w}^{k+1}) - L_\rho(\mathbf{V}^k, \mathbf{U}^k, \mathbf{s}^k, \mathbf{w}^k)$ converges to 0, $\|\mathbf{V}^{k+1} - \mathbf{V}^k\|_F$, $\|\mathbf{U}^{k+1} - \mathbf{U}^k\|_F$, and $\|\mathbf{s}^{k+1} - \mathbf{s}^k\|_2$ also converge to 0. Finally, from Lemma A.4, we have:

$$L_\rho(\mathbf{V}^{k+1}, \mathbf{U}^{k+1}, \mathbf{s}^{k+1}, \mathbf{w}^{k+1}) - L_\rho(\mathbf{V}^{k+1}, \mathbf{U}^{k+1}, \mathbf{s}^{k+1}, \mathbf{w}^k) = \rho \|\mathbf{w}^{k+1} - \mathbf{w}^k\|_2^2$$

$$\leq \frac{3\lambda_{ex}^2 C_V^2}{\rho} \|\mathbf{s}^{k+1} - \mathbf{s}^k\|_2^2 + \frac{6\lambda_{ex}^2 C_V C_s}{\rho} \|\mathbf{V}^{k+1} - \mathbf{V}^k\|_F^2,$$

and then we can also state that $\|\mathbf{w}^{k+1} - \mathbf{w}^k\|_2$ converges to 0.

We next prove the second part of the theorem. Since $\mathbf{V}^{k+1}$ minimizes $L_\rho(\mathbf{V}, \mathbf{U}^k, \mathbf{s}^k, \mathbf{w}^k)$, it holds that $\nabla_{\mathbf{V}} L_\rho(\mathbf{V}^{k+1}, \mathbf{U}^k, \mathbf{s}^k, \mathbf{w}^k) = 0$, and we obtain the following inequality from Lemma A.1:

$$\|\nabla_{\mathbf{V}} L_\rho(\mathbf{V}^k, \mathbf{U}^k, \mathbf{s}^k, \mathbf{w}^k)\|_F = \|\nabla_{\mathbf{V}} L_\rho(\mathbf{V}^{k+1}, \mathbf{U}^k, \mathbf{s}^k, \mathbf{w}^k) - \nabla_{\mathbf{V}} L_\rho(\mathbf{V}^k, \mathbf{U}^k, \mathbf{s}^k, \mathbf{w}^k)\|_F$$

$$= \|\nabla_{\mathbf{V}} g(\mathbf{V}^{k+1}, \mathbf{U}^k, \mathbf{s}^k) - \nabla_{\mathbf{V}} g(\mathbf{V}^k, \mathbf{U}^k, \mathbf{s}^k)\|_F$$

$$\leq \sqrt{|\mathcal{V}|} \left( (1 + \alpha_0) C_U + \lambda_{ex} C_s + \bar{\lambda}_V \right) \|\mathbf{V}^{k+1} - \mathbf{V}^k\|_F.$$

Since $\|\mathbf{V}^{k+1} - \mathbf{V}^k\|_F$ converges to 0, $\nabla_{\mathbf{V}} L_\rho(\mathbf{V}^k, \mathbf{U}^k, \mathbf{s}^k, \mathbf{w}^k)$ converge to 0. Similarly, we have:

$$\|\nabla_{\mathbf{U}} L_\rho(\mathbf{V}^k, \mathbf{U}^k, \mathbf{s}^k, \mathbf{w}^k)\|_F$$

$$= \left\| \nabla_{\mathbf{U}} g(\mathbf{V}^k, \mathbf{U}^k, \mathbf{s}^k) + \frac{\rho}{|\mathcal{U}|^2} \mathbb{1}\mathbb{1}^\top \mathbf{U}^k + \frac{\rho}{|\mathcal{U}|} \mathbb{1}(\mathbf{w}^k - \mathbf{s}^k)^\top \right\|_F$$

$$= \left\| \nabla_{\mathbf{U}} g(\mathbf{V}^k, \mathbf{U}^k, \mathbf{s}^k) - \nabla_{\mathbf{U}} g(\mathbf{V}^k, \mathbf{U}^{k-1}, \mathbf{s}^{k-1}) + \frac{\rho}{|\mathcal{U}|} \mathbb{1}(\mathbf{w}^k - \mathbf{w}^{k-1})^\top - \frac{\rho}{|\mathcal{U}|} \mathbb{1}(\mathbf{s}^k - \mathbf{s}^{k-1})^\top - \frac{1}{\gamma}(\mathbf{U}^k - \mathbf{U}^{k-1}) \right\|_F$$

$$\leq \|\nabla_{\mathbf{U}} g(\mathbf{V}^k, \mathbf{U}^k, \mathbf{s}^k) - \nabla_{\mathbf{U}} g(\mathbf{V}^k, \mathbf{U}^{k-1}, \mathbf{s}^{k-1})\|_F + \frac{1}{\gamma} \|\mathbf{U}^k - \mathbf{U}^{k-1}\|_F + \frac{\rho}{|\mathcal{U}|} \|\mathbb{1}(\mathbf{w}^k - \mathbf{w}^{k-1})^\top\|_F + \frac{\rho}{|\mathcal{U}|} \|\mathbb{1}(\mathbf{s}^k - \mathbf{s}^{k-1})^\top\|_F$$

$$\leq \sqrt{|\mathcal{U}|} \left( (1 + \alpha_0) C_V + \bar{\lambda}_U \right) \|\mathbf{U}^k - \mathbf{U}^{k-1}\|_F + \frac{1}{\gamma} \|\mathbf{U}^k - \mathbf{U}^{k-1}\|_F + \frac{\rho}{|\mathcal{U}|} \|\mathbb{1}(\mathbf{w}^k - \mathbf{w}^{k-1})^\top\|_F + \frac{\rho}{|\mathcal{U}|} \|\mathbb{1}(\mathbf{s}^k - \mathbf{s}^{k-1})^\top\|_F,$$

where the second equality follows from the fact that $\mathbf{U}^k$ minimizes $\frac{\rho}{2} \|\frac{1}{|\mathcal{U}|} \mathbf{U}^\top \mathbb{1} + \mathbf{w}^{k-1} - \mathbf{s}^{k-1}\|_2^2 - \frac{\rho}{2} \|\mathbf{w}^{k-1}\|_2^2 + \frac{1}{2\gamma} \|\mathbf{U} - \mathbf{U}^{k-1}\|_F^2 + \langle \mathbf{U} - \mathbf{U}^{k-1}, \nabla_{\mathbf{U}} g(\mathbf{V}^k, \mathbf{U}^{k-1}, \mathbf{s}^{k-1})\rangle_F$. Since $\|\mathbf{U}^k - \mathbf{U}^{k-1}\|_F$, $\|\mathbf{w}^k - \mathbf{w}^{k-1}\|_2$ converge to 0, this inequality implies that $\nabla_{\mathbf{U}} L_\rho(\mathbf{V}^k, \mathbf{U}^k, \mathbf{s}^k, \mathbf{w}^k)$ converges to 0.

Because $\mathbf{s}^k$ minimizes $L_\rho(\mathbf{V}^k, \mathbf{U}^k, \mathbf{s}, \mathbf{w}^{k-1})$, we also have

$$\|\nabla_{\mathbf{s}} L_\rho(\mathbf{V}^k, \mathbf{U}^k, \mathbf{s}^k, \mathbf{w}^k)\|_2 = \|\nabla_{\mathbf{s}} L_\rho(\mathbf{V}^k, \mathbf{U}^k, \mathbf{s}^k, \mathbf{w}^{k-1}) - \nabla_{\mathbf{s}} L_\rho(\mathbf{V}^k, \mathbf{U}^k, \mathbf{s}^k, \mathbf{w}^k)\|_2$$

$$= \rho \|\mathbf{w}^k - \mathbf{w}^{k-1}\|_2.$$

Thus, since $\|\mathbf{w}^k - \mathbf{w}^{k-1}\|_2$ converges to 0, $\nabla_{\mathbf{s}} L_\rho(\mathbf{V}^k, \mathbf{U}^k, \mathbf{s}^k, \mathbf{w}^k)$ converges to 0.

Finally, it holds that

$$\|\nabla_{\mathbf{w}} L_\rho(\mathbf{V}^k, \mathbf{U}^k, \mathbf{s}^k, \mathbf{w}^k)\|_2 = \rho \left\| \frac{1}{|\mathcal{U}|} (\mathbf{U}^k)^\top \mathbb{1} - \mathbf{s}^k \right\|_2$$

$$= \rho \|\mathbf{w}^k - \mathbf{w}^{k-1}\|_2,$$

and hence $\nabla_{\mathbf{w}} L_\rho(\mathbf{V}^k, \mathbf{U}^k, \mathbf{s}^k, \mathbf{w}^k)$ converges to 0. □

## A.2 Proof of Lemma A.2

Proof. From the definition of $L_\rho(\mathbf{V}, \mathbf{U}, \mathbf{s}^k, \mathbf{w}^k)$, we have:

$$L_\rho(\mathbf{V}^{k+1}, \mathbf{U}^{k+1}, \mathbf{s}^k, \mathbf{w}^k) - L_\rho(\mathbf{V}^k, \mathbf{U}^k, \mathbf{s}^k, \mathbf{w}^k)$$

$$= g(\mathbf{V}^{k+1}, \mathbf{U}^{k+1}, \mathbf{s}^k) + \frac{\rho}{2} \left\| \mathbf{w}^k + \frac{1}{|\mathcal{U}|} (\mathbf{U}^{k+1})^\top \mathbb{1} - \mathbf{s}^k \right\|_2^2 - \frac{\rho}{2} \|\mathbf{w}^k\|_2^2 - g(\mathbf{V}^k, \mathbf{U}^k, \mathbf{s}^k) - \frac{\rho}{2} \left\| \mathbf{w}^k + \frac{1}{|\mathcal{U}|} (\mathbf{U}^k)^\top \mathbb{1} - \mathbf{s}^k \right\|_2^2 + \frac{\rho}{2} \|\mathbf{w}^k\|_2^2$$

$$= g(\mathbf{V}^{k+1}, \mathbf{U}^{k+1}, \mathbf{s}^k) - g(\mathbf{V}^{k+1}, \mathbf{U}^k, \mathbf{s}^k) + g(\mathbf{V}^{k+1}, \mathbf{U}^k, \mathbf{s}^k) - g(\mathbf{V}^k, \mathbf{U}^k, \mathbf{s}^k) + \frac{\rho}{2} \left\| \mathbf{w}^k + \frac{1}{|\mathcal{U}|} (\mathbf{U}^{k+1})^\top \mathbb{1} - \mathbf{s}^k \right\|_2^2 - \frac{\rho}{2} \left\| \mathbf{w}^k + \frac{1}{|\mathcal{U}|} (\mathbf{U}^k)^\top \mathbb{1} - \mathbf{s}^k \right\|_2^2.$$

$$(9)$$

Denoting the Gram matrix of $\mathbf{U}$ by $\mathbf{G}_U = \mathbf{U}^\top \mathbf{U}$, we have

$$\langle \nabla_\mathbf{V} g(\mathbf{V}, \mathbf{U}, \mathbf{s}) - \nabla_\mathbf{V} g(\mathbf{V}', \mathbf{U}, \mathbf{s}), \mathbf{V} - \mathbf{V}' \rangle_F$$

$$= \sum_{j \in \mathcal{V}} \left\langle \left( \left( \sum_{i \in \mathcal{U}} r_{i,j} \mathbf{u}_i \mathbf{u}_i^\top + \alpha_0 \mathbf{G}_U + \lambda_{ex} \mathbf{s}\mathbf{s}^\top + \lambda_V^{(j)} \mathbf{I} \right) \mathbf{v}_j - \sum_{i \in \mathcal{U}} r_{i,j} \mathbf{u}_i - \left( \sum_{i \in \mathcal{U}} r_{i,j} \mathbf{u}_i \mathbf{u}_i^\top + \alpha_0 \mathbf{G}_U + \lambda_{ex} \mathbf{s}\mathbf{s}^\top + \lambda_V^{(j)} \mathbf{I} \right) \mathbf{v}_j' + \sum_{i \in \mathcal{U}} r_{i,j} \mathbf{u}_i, \mathbf{v}_j - \mathbf{v}_j' \right\rangle$$

$$= \sum_{j \in \mathcal{V}} \left\langle \left( \sum_{i \in \mathcal{U}} r_{i,j} \mathbf{u}_i \mathbf{u}_i^\top \right) (\mathbf{v}_j - \mathbf{v}_j'), \mathbf{v}_j - \mathbf{v}_j' \right\rangle + \sum_{j \in \mathcal{V}} \left\langle \alpha_0 \mathbf{G}_U (\mathbf{v}_j - \mathbf{v}_j'), \mathbf{v}_j - \mathbf{v}_j' \right\rangle$$

$$+ \sum_{j \in \mathcal{V}} \left\langle \lambda_{ex} \mathbf{s}\mathbf{s}^\top (\mathbf{v}_j - \mathbf{v}_j'), \mathbf{v}_j - \mathbf{v}_j' \right\rangle + \sum_{j \in \mathcal{V}} \left\langle \lambda_V^{(j)} (\mathbf{v}_j - \mathbf{v}_j'), \mathbf{v}_j - \mathbf{v}_j' \right\rangle$$

$$= \sum_{j \in \mathcal{V}} \sum_{i \in \mathcal{U}} r_{i,j} (\mathbf{u}_i^\top (\mathbf{v}_j - \mathbf{v}_j'))^2 + \alpha_0 \sum_{j \in \mathcal{V}} \sum_{i \in \mathcal{U}} (\mathbf{u}_i^\top (\mathbf{v}_j - \mathbf{v}_j'))^2 + \lambda_{ex} \sum_{j \in \mathcal{V}} (\mathbf{s}^\top (\mathbf{v}_j - \mathbf{v}_j'))^2 + \sum_{j \in \mathcal{V}} \| \lambda_V^{(j)} (\mathbf{v}_j - \mathbf{v}_j') \|_2^2$$

$$\geq \underline{\lambda}_V \sum_{j \in \mathcal{V}} \| \mathbf{v}_j - \mathbf{v}_j' \|_2^2 = \underline{\lambda}_V \| \mathbf{V} - \mathbf{V}' \|_F^2,$$

where $\underline{\lambda}_V = \min_{j \in \mathcal{V}} \lambda_V^{(j)}$. Thus, the function $g$ is $\underline{\lambda}_V$-strongly convex with respect to $\mathbf{V}$. We also have

$$g(\mathbf{V}^{k+1}, \mathbf{U}^k, \mathbf{s}^k) - g(\mathbf{V}^k, \mathbf{U}^k, \mathbf{s}^k) \leq \langle \nabla_\mathbf{V} g(\mathbf{V}^{k+1}, \mathbf{U}^k, \mathbf{s}^k), \mathbf{V}^{k+1} - \mathbf{V}^k \rangle_F - \frac{\underline{\lambda}_V}{2} \| \mathbf{V}^{k+1} - \mathbf{V}^k \|_F^2$$

$$= -\frac{\underline{\lambda}_V}{2} \| \mathbf{V}^{k+1} - \mathbf{V}^k \|_F^2, \tag{10}$$

where the last equality follows from the fact that $\mathbf{V}^{k+1}$ minimizes $g(\mathbf{V}, \mathbf{U}^k, \mathbf{s}^k)$; hence $\nabla_\mathbf{V} g(\mathbf{V}^{k+1}, \mathbf{U}^k, \mathbf{s}^k) = 0$ holds. Moreover, since $\mathbf{U}^{k+1}$ minimizes $\frac{\rho}{2} \left\| \mathbf{w}^k + \frac{1}{|\mathcal{U}|} (\mathbf{U})^\top \mathbb{1} - \mathbf{s}^k \right\|_2^2 - \frac{\rho}{2} \| \mathbf{w}^k \|_2^2 + \frac{1}{2\gamma} \| \mathbf{U} - \mathbf{U}^k \|_F^2 + \langle \mathbf{U} - \mathbf{U}^k, \nabla_\mathbf{U} g(\mathbf{V}^{k+1}, \mathbf{U}^k, \mathbf{s}^k) \rangle_F$, we have:

$$\frac{\rho}{2} \left\| \mathbf{w}^k + \frac{1}{|\mathcal{U}|} (\mathbf{U}^{k+1})^\top \mathbb{1} - \mathbf{s}^k \right\|_2^2 - \frac{\rho}{2} \| \mathbf{w}^k \|_2^2 + \frac{1}{2\gamma} \| \mathbf{U}^{k+1} - \mathbf{U}^k \|_F^2 + \langle \mathbf{U}^{k+1} - \mathbf{U}^k, \nabla_\mathbf{U} g(\mathbf{V}^{k+1}, \mathbf{U}^k, \mathbf{s}^k) \rangle_F$$

$$\leq \frac{\rho}{2} \left\| \mathbf{w}^k + \frac{1}{|\mathcal{U}|} (\mathbf{U}^k)^\top \mathbb{1} - \mathbf{s}^k \right\|_2^2 - \frac{\rho}{2} \| \mathbf{w}^k \|_2^2. \tag{11}$$

By combining Eqs. (9) to (11), we obtain:

$$L_\rho(\mathbf{V}^{k+1}, \mathbf{U}^{k+1}, \mathbf{s}^k, \mathbf{w}^k) - L_\rho(\mathbf{V}^k, \mathbf{U}^k, \mathbf{s}^k, \mathbf{w}^k)$$

$$\leq g(\mathbf{V}^{k+1}, \mathbf{U}^{k+1}, \mathbf{s}^k) - g(\mathbf{V}^{k+1}, \mathbf{U}^k, \mathbf{s}^k) - \langle \mathbf{U}^{k+1} - \mathbf{U}^k, \nabla_\mathbf{U} g(\mathbf{V}^{k+1}, \mathbf{U}^k, \mathbf{s}^k) \rangle_F - \frac{1}{2\gamma} \| \mathbf{U}^{k+1} - \mathbf{U}^k \|_F^2 - \frac{\underline{\lambda}_V}{2} \| \mathbf{V}^{k+1} - \mathbf{V}^k \|_F^2. \tag{12}$$

On the other hand, under the assumption in Theorem 3.1, from Lemma A.1, the function $g(\mathbf{V}^{k+1}, \mathbf{U}, \mathbf{s}^k)$ is $\sqrt{|\mathcal{U}|} ((1 + \alpha_0) C_V + \bar{\lambda}_U)$-smooth with respect to $\mathbf{U}$. Then, for any $\mathbf{U}, \mathbf{U}'$, we have:

$$g(\mathbf{V}^{k+1}, \mathbf{U}', \mathbf{s}^k) - g(\mathbf{V}^{k+1}, \mathbf{U}, \mathbf{s}^k) - \langle \nabla_\mathbf{U} g(\mathbf{V}^{k+1}, \mathbf{U}, \mathbf{s}^k), \mathbf{U}' - \mathbf{U} \rangle_F \leq \frac{\sqrt{|\mathcal{U}|} ((1 + \alpha_0) C_V + \bar{\lambda}_U)}{2} \| \mathbf{U} - \mathbf{U}' \|_F^2. \tag{13}$$

By combining Eq. (12) and Eq. (13), we obtain the following inequality:

$$L_\rho(\mathbf{V}^{k+1}, \mathbf{U}^{k+1}, \mathbf{s}^k, \mathbf{w}^k) - L_\rho(\mathbf{V}^k, \mathbf{U}^k, \mathbf{s}^k, \mathbf{w}^k) \leq \frac{\sqrt{|\mathcal{U}|} ((1 + \alpha_0) C_V + \bar{\lambda}_U) - 1/\gamma}{2} \| \mathbf{U}^{k+1} - \mathbf{U}^k \|_F^2 - \frac{\underline{\lambda}_V}{2} \| \mathbf{V}^{k+1} - \mathbf{V}^k \|_F^2.$$

□

## A.3 Proof of Lemma A.3

Proof. Let us define $h^k(\mathbf{s}) = g(\mathbf{V}^{k+1}, \mathbf{U}^{k+1}, \mathbf{s}) + \frac{\rho}{2} \left\| \mathbf{w}^k + \frac{1}{|\mathcal{U}|} (\mathbf{U}^{k+1})^\top \mathbb{1} - \mathbf{s} \right\|_2^2 - \frac{\rho}{2} \| \mathbf{w}^k \|_2^2$. We have:

$$\langle \nabla h^k(\mathbf{s}) - \nabla h^k(\mathbf{s}'), \mathbf{s} - \mathbf{s}' \rangle$$

$$= \left\langle \nabla_\mathbf{s} g(\mathbf{V}^{k+1}, \mathbf{U}^{k+1}, \mathbf{s}) - \rho \left( \mathbf{w}^k + \frac{1}{|\mathcal{U}|} (\mathbf{U}^{k+1})^\top \mathbb{1} - \mathbf{s} \right) - \nabla_\mathbf{s} g(\mathbf{V}^{k+1}, \mathbf{U}^{k+1}, \mathbf{s}') + \rho \left( \mathbf{w}^k + \frac{1}{|\mathcal{U}|} (\mathbf{U}^{k+1})^\top \mathbb{1} - \mathbf{s}' \right), \mathbf{s} - \mathbf{s}' \right\rangle$$

$$= \left\langle \nabla_\mathbf{s} g(\mathbf{V}^{k+1}, \mathbf{U}^{k+1}, \mathbf{s}) - \nabla_\mathbf{s} g(\mathbf{V}^{k+1}, \mathbf{U}^{k+1}, \mathbf{s}') + \rho(\mathbf{s} - \mathbf{s}'), \mathbf{s} - \mathbf{s}' \right\rangle$$

$$\geq \rho \| \mathbf{s} - \mathbf{s}' \|_2^2,$$

where the inequality follows from the convexity of $g(\mathbf{V}^{k+1}, \mathbf{U}^{k+1}, \cdot)$. Consequently, $h^k$ is a $\rho$-strongly convex function. Therefore, we have:

$$L_\rho(\mathbf{V}^{k+1}, \mathbf{U}^{k+1}, \mathbf{s}^{k+1}, \mathbf{w}^k) - L_\rho(\mathbf{V}^{k+1}, \mathbf{U}^{k+1}, \mathbf{s}^k, \mathbf{w}^k)$$

$$= h^k(\mathbf{s}^{k+1}) - h^k(\mathbf{s}^k) \le \langle \nabla h^k(\mathbf{s}^{k+1}), \mathbf{s}^{k+1} - \mathbf{s}^k \rangle - \frac{\rho}{2} \|\mathbf{s}^{k+1} - \mathbf{s}^k\|_2^2 = -\frac{\rho}{2} \|\mathbf{s}^{k+1} - \mathbf{s}^k\|_2^2,$$

where the last equality follows from that $\mathbf{s}^{k+1}$ minimizes $h^k(\mathbf{s})$, i.e., $\nabla h^k(\mathbf{s}^{k+1}) = 0$. □

## A.4 Proof of Lemma A.4

PROOF. From the definition of $L_\rho(\mathbf{V}^{k+1}, \mathbf{U}^{k+1}, \mathbf{s}^{k+1}, \mathbf{w})$ and the update rule of $\mathbf{w}^k$, we have:

$$L_\rho(\mathbf{V}^{k+1}, \mathbf{U}^{k+1}, \mathbf{s}^{k+1}, \mathbf{w}^{k+1}) - L_\rho(\mathbf{V}^{k+1}, \mathbf{U}^{k+1}, \mathbf{s}^{k+1}, \mathbf{w}^k)$$

$$= \frac{\rho}{2} \left\| \mathbf{w}^{k+1} + \frac{1}{|\mathcal{U}|} (\mathbf{U}^{k+1})^\top \mathbb{1} - \mathbf{s}^{k+1} \right\|_2^2 - \frac{\rho}{2} \|\mathbf{w}^{k+1}\|_2^2 - \frac{\rho}{2} \left\| \mathbf{w}^k + \frac{1}{|\mathcal{U}|} (\mathbf{U}^{k+1})^\top \mathbb{1} - \mathbf{s}^{k+1} \right\|_2^2 + \frac{\rho}{2} \|\mathbf{w}^k\|_2^2$$

$$= \rho \langle \mathbf{w}^{k+1} - \mathbf{w}^k, \frac{1}{|\mathcal{U}|} (\mathbf{U}^{k+1})^\top \mathbb{1} - \mathbf{s}^{k+1} \rangle = \rho \|\mathbf{w}^{k+1} - \mathbf{w}^k\|_2^2. \tag{14}$$

On the other hand, since $\mathbf{s}^{k+1}$ minimizes the convex function $h^k(\mathbf{s})$, the first-order optimality condition implies:

$$\nabla h^k(\mathbf{s}^{k+1}) = \nabla_s g(\mathbf{V}^{k+1}, \mathbf{U}^{k+1}, \mathbf{s}^{k+1}) - \rho \left( \mathbf{w}^k + \frac{1}{|\mathcal{U}|} (\mathbf{U}^{k+1})^\top \mathbb{1} - \mathbf{s}^{k+1} \right)$$

$$= \nabla_s g(\mathbf{V}^{k+1}, \mathbf{U}^{k+1}, \mathbf{s}^{k+1}) - \rho \mathbf{w}^{k+1} = 0.$$

Thus, it holds that

$$\mathbf{w}^{k+1} = \frac{1}{\rho} \nabla_s g(\mathbf{V}^{k+1}, \mathbf{U}^{k+1}, \mathbf{s}^{k+1}). \tag{15}$$

By combining Eq. (14), Eq. (15), and Lemma A.1, we obtain:

$$L_\rho(\mathbf{V}^{k+1}, \mathbf{U}^{k+1}, \mathbf{s}^{k+1}, \mathbf{w}^{k+1}) - L_\rho(\mathbf{V}^{k+1}, \mathbf{U}^{k+1}, \mathbf{s}^{k+1}, \mathbf{w}^k)$$

$$= \frac{1}{\rho} \|\nabla_s g(\mathbf{V}^{k+1}, \mathbf{U}^{k+1}, \mathbf{s}^{k+1}) - \nabla_s g(\mathbf{V}^k, \mathbf{U}^k, \mathbf{s}^k)\|_2^2$$

$$= \frac{1}{\rho} \|\nabla_s g(\mathbf{V}^{k+1}, \mathbf{U}^{k+1}, \mathbf{s}^{k+1}) - \nabla_s g(\mathbf{V}^{k+1}, \mathbf{U}^k, \mathbf{s}^k) + \nabla_s g(\mathbf{V}^{k+1}, \mathbf{U}^k, \mathbf{s}^k) - \nabla_s g(\mathbf{V}^k, \mathbf{U}^k, \mathbf{s}^k)\|_2^2$$

$$\le \frac{1}{\rho} \left( \|\nabla_s g(\mathbf{V}^{k+1}, \mathbf{U}^{k+1}, \mathbf{s}^{k+1}) - \nabla_s g(\mathbf{V}^{k+1}, \mathbf{U}^k, \mathbf{s}^k)\|_2 + \|\nabla_s g(\mathbf{V}^{k+1}, \mathbf{U}^k, \mathbf{s}^k) - \nabla_s g(\mathbf{V}^k, \mathbf{U}^k, \mathbf{s}^k)\|_2 \right)^2$$

$$\le \frac{1}{\rho} \left( \lambda_{ex} \|\mathbf{V}^{k+1}\|_F^2 \|\mathbf{s}^{k+1} - \mathbf{s}^k\|_2 + \lambda_{ex} \left( \|\mathbf{V}^{k+1}\|_F + \|\mathbf{V}^k\|_F \right) \|\mathbf{V}^{k+1} - \mathbf{V}^k\|_F \|\mathbf{s}^k\|_2 \right)^2$$

$$\le \frac{3}{\rho} \left( \lambda_{ex}^2 \|\mathbf{V}^{k+1}\|_F^4 \|\mathbf{s}^{k+1} - \mathbf{s}^k\|_2^2 + \lambda_{ex}^2 \left( \|\mathbf{V}^{k+1}\|_F^2 + \|\mathbf{V}^k\|_F^2 \right) \|\mathbf{V}^{k+1} - \mathbf{V}^k\|_F^2 \|\mathbf{s}^k\|_2^2 \right)$$

$$\le \frac{3\lambda_{ex}^2}{\rho} \left( C_V^2 \|\mathbf{s}^{k+1} - \mathbf{s}^k\|_2^2 + 2 C_V C_s \|\mathbf{V}^{k+1} - \mathbf{V}^k\|_F^2 \right),$$

where the third inequality follows from $(a + b + c)^2 \le 3(a^2 + b^2 + c^2)$ for $a, b, c \in \mathbb{R}$. □

## A.5 Proof of Lemma A.5

PROOF OF LEMMA A.5. Under the assumption in Theorem 3.1, the function $g(\mathbf{V}^k, \mathbf{U}^k, \mathbf{s})$ is $\lambda_{ex} C_V$-smooth with respect to $\mathbf{s}$ from Lemma A.1, and then we have, for any $\mathbf{s}, \mathbf{s}'$,

$$g(\mathbf{V}^k, \mathbf{U}^k, \mathbf{s}') - g(\mathbf{V}^k, \mathbf{U}^k, \mathbf{s}) - \langle \nabla_s g(\mathbf{V}^k, \mathbf{U}^k, \mathbf{s}), \mathbf{s}' - \mathbf{s} \rangle \le \frac{\lambda_{ex} C_V}{2} \|\mathbf{s} - \mathbf{s}'\|_2^2. \tag{16}$$

By combining Eq. (15) and Eq. (16), we obtain:

$$L_\rho(\mathbf{V}^k, \mathbf{U}^k, \mathbf{s}^k, \mathbf{w}^k) = g(\mathbf{V}^k, \mathbf{U}^k, \mathbf{s}^k) + \frac{\rho}{2}\left\|\mathbf{w}^k + \frac{1}{|\mathcal{U}|}(\mathbf{U}^k)^\top \mathbb{1} - \mathbf{s}^k\right\|_2^2 - \frac{\rho}{2}\|\mathbf{w}^k\|_2^2$$

$$= g(\mathbf{V}^k, \mathbf{U}^k, \mathbf{s}^k) + \rho\langle \mathbf{w}^k, \frac{1}{|\mathcal{U}|}(\mathbf{U}^k)^\top \mathbb{1} - \mathbf{s}^k\rangle + \frac{\rho}{2}\left\|\frac{1}{|\mathcal{U}|}(\mathbf{U}^k)^\top \mathbb{1} - \mathbf{s}^k\right\|_2^2$$

$$= g(\mathbf{V}^k, \mathbf{U}^k, \mathbf{s}^k) - \langle \nabla_{\mathbf{s}} g(\mathbf{V}^k, \mathbf{U}^k, \mathbf{s}^k), \mathbf{s}^k - \frac{1}{|\mathcal{U}|}(\mathbf{U}^k)^\top \mathbb{1}\rangle + \frac{\rho}{2}\left\|\frac{1}{|\mathcal{U}|}(\mathbf{U}^k)^\top \mathbb{1} - \mathbf{s}^k\right\|_2^2$$

$$\geq g(\mathbf{V}^k, \mathbf{U}^k, \frac{1}{|\mathcal{U}|}(\mathbf{U}^k)^\top \mathbb{1}) + \frac{\rho - \lambda_{ex} C_V}{2}\left\|\frac{1}{|\mathcal{U}|}(\mathbf{U}^k)^\top \mathbb{1} - \mathbf{s}^k\right\|_2^2$$

$$\geq \frac{\rho - \lambda_{ex} C_V}{2}\left\|\frac{1}{|\mathcal{U}|}(\mathbf{U}^k)^\top \mathbb{1} - \mathbf{s}^k\right\|_2^2,$$

where the last inequality follows from $g(\mathbf{V}, \mathbf{U}, \mathbf{s}) \geq 0$ for any $\mathbf{v}$, $\mathbf{U}$, and $\mathbf{s}$. □

## A.6 Proof for Lemma A.1

PROOF. For fixed $\mathbf{U}$, $\mathbf{s}$, for all $\mathbf{V}$, $\mathbf{V}'$, we have the following

$$\|\nabla_{\mathbf{V}} g(\mathbf{V}, \mathbf{U}, \mathbf{s}) - \nabla_{\mathbf{V}} g(\mathbf{V}', \mathbf{U}, \mathbf{s})\|_F$$

$$= \sum_{j\in\mathcal{V}}\left\|\left(\sum_{i\in\mathcal{U}} r_{i,j}\mathbf{u}_i\mathbf{u}_i^\top + \alpha_0 \mathbf{G}_U + \lambda_{ex}\mathbf{s}\mathbf{s}^\top + \lambda_V^{(j)}\mathbf{I}\right)(\mathbf{v}_j - \mathbf{v}_j')\right\|_2$$

$$\leq \sum_{j\in\mathcal{V}}\sum_{i\in\mathcal{U}}\|r_{i,j}\mathbf{u}_i\mathbf{u}_i^\top(\mathbf{v}_j - \mathbf{v}_j')\|_2 + \sum_{j\in\mathcal{V}}\sum_{i\in\mathcal{U}}\|\alpha_0\mathbf{u}_i\mathbf{u}_i^\top(\mathbf{v}_j - \mathbf{v}_j')\|_2 + \sum_{j\in\mathcal{V}}\|\lambda_{ex}\mathbf{s}\mathbf{s}^\top(\mathbf{v}_j - \mathbf{v}_j')\|_2 + \sum_{j\in\mathcal{V}}\|\lambda_V^{(j)}(\mathbf{v}_j - \mathbf{v}_j')\|_2$$

$$= \sum_{j\in\mathcal{V}}\sum_{i\in\mathcal{U}} r_{i,j}|\mathbf{u}_i^\top(\mathbf{v}_j - \mathbf{v}_j')|\cdot\|\mathbf{u}_i\|_2 + \alpha_0\sum_{j\in\mathcal{V}}\sum_{i\in\mathcal{U}}|\mathbf{u}_i^\top(\mathbf{v}_j - \mathbf{v}_j')|\cdot\|\mathbf{u}_i\|_2 + \lambda_{ex}\sum_{j\in\mathcal{V}}|\mathbf{s}^\top(\mathbf{v}_j - \mathbf{v}_j')|\cdot\|\mathbf{s}\|_2 + \sum_{j\in\mathcal{V}}\|\lambda_V^{(j)}(\mathbf{v}_j - \mathbf{v}_j')\|_2$$

$$\leq \sum_{j\in\mathcal{V}}\sum_{i\in\mathcal{U}} r_{i,j}\|\mathbf{u}_i\|_2^2\|\mathbf{v}_j - \mathbf{v}_j'\|_2 + \alpha_0\sum_{j\in\mathcal{V}}\sum_{i\in\mathcal{U}}\|\mathbf{u}_i\|_2^2\|\mathbf{v}_j - \mathbf{v}_j'\|_2 + \lambda_{ex}\sum_{j\in\mathcal{V}}\|\mathbf{s}\|_2^2\|\mathbf{v}_j - \mathbf{v}_j'\|_2 + \bar{\lambda}_V\sum_{j\in\mathcal{V}}\|\mathbf{v}_j - \mathbf{v}_j'\|_2$$

$$\leq \left((1+\alpha_0)\sum_{i\in\mathcal{U}}\|\mathbf{u}_i\|_2^2 + \lambda_{ex}\|\mathbf{s}\|_2^2 + \bar{\lambda}_V\right)\sum_{j\in\mathcal{V}}\|\mathbf{v}_j - \mathbf{v}_j'\|_2$$

$$\leq \sqrt{|\mathcal{V}|}\left((1+\alpha_0)\|\mathbf{U}\|_F^2 + \lambda_{ex}\|\mathbf{s}\|_2^2 + \bar{\lambda}_V\right)\|\mathbf{V} - \mathbf{V}'\|_F,$$

where $\bar{\lambda}_V = \max_{j\in\mathcal{V}}\lambda_V^{(j)}$. Note that the second inequality follows from the Cauchy–Schwarz inequality.

In addition, we have, for a fixed $\mathbf{V}$, for all $\mathbf{s}$, $\mathbf{s}'$ and $\mathbf{U}$, $\mathbf{U}'$,

$$\|\nabla_{\mathbf{U}} g(\mathbf{V}, \mathbf{U}, \mathbf{s}) - \nabla_{\mathbf{U}} g(\mathbf{V}, \mathbf{U}', \mathbf{s}')\|_F$$

$$= \sum_{i\in\mathcal{U}}\left\|\left(\sum_{j\in\mathcal{V}} r_{i,j}\mathbf{v}_j\mathbf{v}_j^\top + \alpha_0\sum_{j\in\mathcal{V}}\mathbf{v}_j\mathbf{v}_j^\top + \lambda_U^{(i)}\mathbf{I}\right)\mathbf{u}_i - \sum_{j\in\mathcal{V}} r_{i,j}\mathbf{v}_j - \left(\sum_{j\in\mathcal{V}} r_{i,j}\mathbf{v}_j\mathbf{v}_j^\top + \alpha_0\sum_{j\in\mathcal{V}}\mathbf{v}_j\mathbf{v}_j^\top + \lambda_U^{(i)}\mathbf{I}\right)\mathbf{u}_i' + \sum_{j\in\mathcal{V}} r_{i,j}\mathbf{v}_j\right\|_2$$

$$= \sum_{i\in\mathcal{U}}\left\|\left(\sum_{j\in\mathcal{V}} r_{i,j}\mathbf{v}_j\mathbf{v}_j^\top + \alpha_0\sum_{j\in\mathcal{V}}\mathbf{v}_j\mathbf{v}_j^\top + \lambda_U^{(i)}\mathbf{I}\right)(\mathbf{u}_i - \mathbf{u}_i')\right\|_2$$

$$\leq \sum_{i\in\mathcal{U}}\sum_{j\in\mathcal{V}}\left\|\left(r_{i,j}\mathbf{v}_j\mathbf{v}_j^\top\right)(\mathbf{u}_i - \mathbf{u}_i')\right\|_2 + \sum_{i\in\mathcal{U}}\sum_{j\in\mathcal{V}}\left\|\left(\alpha_0\mathbf{v}_j\mathbf{v}_j^\top\right)(\mathbf{u}_i - \mathbf{u}_i')\right\| + \sum_{i\in\mathcal{U}}\left\|\lambda_U^{(i)}(\mathbf{u}_i - \mathbf{u}_i')\right\|_2$$

$$= \sum_{i\in\mathcal{U}}\sum_{j\in\mathcal{V}} r_{i,j}\left\|\mathbf{v}_j\mathbf{v}_j^\top(\mathbf{u}_i - \mathbf{u}_i')\right\|_2 + \alpha_0\sum_{i\in\mathcal{U}}\sum_{j\in\mathcal{V}}\left\|\mathbf{v}_j\mathbf{v}_j^\top(\mathbf{u}_i - \mathbf{u}_i')\right\|_2 + \sum_{i\in\mathcal{U}}\left\|\lambda_U^{(i)}(\mathbf{u}_i - \mathbf{u}_i')\right\|_2$$

$$= \sum_{i\in\mathcal{U}}\sum_{j\in\mathcal{V}} r_{i,j}|\mathbf{v}_j^\top(\mathbf{u}_i - \mathbf{u}_i')|\left\|\mathbf{v}_j\right\|_2 + \alpha_0\sum_{i\in\mathcal{U}}\sum_{j\in\mathcal{V}}|\mathbf{v}_j^\top(\mathbf{u}_i - \mathbf{u}_i')|\left\|\mathbf{v}_j\right\|_2 + \sum_{i\in\mathcal{U}}\left\|\lambda_U^{(i)}(\mathbf{u}_i - \mathbf{u}_i')\right\|_2$$

$$\leq \sum_{i\in\mathcal{U}}\sum_{j\in\mathcal{V}} r_{i,j}\left\|\mathbf{v}_j\right\|_2^2\left\|\mathbf{u}_i - \mathbf{u}_i'\right\|_2 + \alpha_0\sum_{i\in\mathcal{U}}\sum_{j\in\mathcal{V}}\left\|\mathbf{v}_j\right\|^2\left\|\mathbf{u}_i - \mathbf{u}_i'\right\|_2 + \sum_{i\in\mathcal{U}}\left\|\lambda_U^{(i)}(\mathbf{u}_i - \mathbf{u}_i')\right\|_2$$

$$\leq (1 + \alpha_0) \left( \sum_{j \in \mathcal{V}} \|\mathbf{v}_j\|_2^2 \right) \sum_{i \in \mathcal{U}} \|\mathbf{u}_i - \mathbf{u}_i'\|_2 + \bar{\lambda}_U \sum_{i \in \mathcal{U}} \|\mathbf{u}_i - \mathbf{u}_i'\|_2$$

$$\leq \sqrt{|\mathcal{U}|} \left( (1 + \alpha_0) \|\mathbf{V}\|_F^2 + \bar{\lambda}_U \right) \|\mathbf{U} - \mathbf{U}'\|_F,$$

where the second inequality follows from the Cauchy–Schwarz inequality.

Here, for a fixed $\mathbf{V}$, for all $\mathbf{s}, \mathbf{s}'$ and $\mathbf{U}, \mathbf{U}'$, we have:

$$\|\nabla_{\mathbf{s}} g(\mathbf{V}, \mathbf{U}, \mathbf{s}) - \nabla_{\mathbf{s}} g(\mathbf{V}, \mathbf{U}', \mathbf{s}')\|_2 = \left\| \lambda_{ex} \mathbf{V}^\top \mathbf{V} \mathbf{s} - \lambda_{ex} \mathbf{V}^\top \mathbf{V} \mathbf{s}' \right\|_2$$

$$= \lambda_{ex} \|\mathbf{V}^\top \mathbf{V}(\mathbf{s} - \mathbf{s}')\|_2$$

$$\leq \lambda_{ex} \|\mathbf{V}^\top \mathbf{V}\|_F \|\mathbf{s} - \mathbf{s}'\|_2$$

$$\leq \lambda_{ex} \|\mathbf{V}\|_F^2 \|\mathbf{s} - \mathbf{s}'\|_2,$$

where the first/second inequality follows from the Cauchy–Schwarz inequality.

Finally, for fixed $\mathbf{U}$ and $\mathbf{s}$, for all $\mathbf{V}, \mathbf{V}'$, we have:

$$\|\nabla_{\mathbf{s}} g(\mathbf{V}, \mathbf{U}, \mathbf{s}) - \nabla_{\mathbf{s}} g(\mathbf{V}', \mathbf{U}, \mathbf{s})\|_2 = \lambda_{ex} \|\mathbf{V}^\top \mathbf{V} \mathbf{s} - \mathbf{V}'^\top \mathbf{V}' \mathbf{s}\|_2$$

$$= \lambda_{ex} \|(\mathbf{V}^\top (\mathbf{V} - \mathbf{V}') + (\mathbf{V} - \mathbf{V}')^\top \mathbf{V}') \mathbf{s}\|_2$$

$$\leq \lambda_{ex} \|\mathbf{V}^\top (\mathbf{V} - \mathbf{V}') \mathbf{s}\|_2 + \lambda_{ex} \|(\mathbf{V} - \mathbf{V}')^\top \mathbf{V}' \mathbf{s}\|_2$$

$$\leq \lambda_{ex} \|\mathbf{V}^\top (\mathbf{V} - \mathbf{V}')\|_F \cdot \|\mathbf{s}\|_2 + \lambda_{ex} \|(\mathbf{V} - \mathbf{V}')^\top \mathbf{V}'\|_F \cdot \|\mathbf{s}\|_2$$

$$\leq \lambda_{ex} (\|\mathbf{V}\|_F + \|\mathbf{V}'\|_F) \cdot \|\mathbf{s}\|_2 \|\mathbf{V} - \mathbf{V}'\|_F,$$

where the second/third inequality follows from the Cauchy–Schwarz inequality. □

