# OpenReview forum: "Scalable and Provably Fair Exposure Control for Large-Scale Recommender Systems"
_ACM.org/TheWebConf/2024/Conference — TheWebConf24_

### Official Review · Reviewer_Rxja · 2023-11-17

**Novelty:** 5
**Technical Quality:** 6

**Review:**

In this paper, the authors proposed a scalable recommendation method for fair exposure. They argued that current fair recommendation methods are not suitable for large-scale scenarios. Hence, they used an iALS method as the backbone and optimized the model using an ADMM-based technique. The scalability issue of recommendation systems is still an open problem in the industry, so I believe the research question in this paper is meaningful. My concern is about the novelty of this paper. although I am not an expert in the field of scalable recommendation systems, I know that iALS and ADMM are quite classical methods to solve the scalability problem of large-scale recommendation. The author should be clearer about how many contributions they have made on top of the classical paradigm. I suggest that the authors expand on the topic of scalable recommendation systems in the related works part.

Pros:
- Interesting and emerging research question.
- A clear and detailed technical illustration presented in a step-by-step manner.

Cons:
- There may be a potential issue with novelty.

**Questions:**

1. Besides taking into account fairness, how does this paper differ from other iALS+ADMM methods?
2. In Table 3, FairRec spent less training time than exADMM, which shows a different pattern compared to the other two datasets. Can the authors explain this phenomenon?

**Reviewer Confidence:**

2: The reviewer is willing to defend the evaluation, but it is likely that the reviewer did not understand parts of the paper

**Scope:**

3: The work is somewhat relevant to the Web and to the track, and is of narrow interest to a sub-community

---

### Official Review · Reviewer_DaC2 · 2023-11-22

**Novelty:** 3
**Technical Quality:** 3

**Review:**

This paper tackles a crucial fairness concern in the domain of Large-Scale Recommender Systems. The authors extend iALS to exADMM from a fairness perspective. In order to accomplish this extension, they successfully address the issue of the fairness regularizer interfering with the separability of optimization subproblems for users and items. Moreover, they offer theoretical assurances of convergence. Through extensive experiments on three recommendation datasets, they demonstrate the effectiveness of exADMM.


Strength:
1. The fairness of Large-Scale Recommender Systems holds significant research value in web applications.
2. The paper introduces exADMM, a model that effectively addresses fairness concerns based on iALS with theoretical guarantees of convergence.
3. Extensive experiments demonstrate that exADMM not only achieves improved fairness outcomes but also exhibits higher computational efficiency.

Weakness:
1. This paper suffers from poor writing and a lack of reader-friendliness, evident in the following aspects:
	Insufficient systematic explanation of related work in the field of Large-Scale Recommender Systems. The paper merely lists a few works related to iALS without clarifying the positioning of this work within the field of Large-Scale Recommender Systems.
	Absence of a toy example to help readers quickly grasp exADMM.
	Some significant typographical errors, such as stating "accuracy" instead of "fairness" in the sentence "In addition to the accuracy metric, we use Gini@K."

2. Some parts of the technical content in this paper are confusing, particularly regarding the regularizer R_ex. The paper initially claims that "The fairness regularizer defined above is the L2 norm of the average scores predicted for the items." However, later on, the paper states, "This is considered one of the reasonable measures of exposure inequality since it takes a large value for items whose average scores are either extremely large or small." It is unclear how this conclusion is reached. If R_ex is defined as the average scores predicted for the items, then it should also be either extremely large or small when the average scores are either extremely large or small.

It should be noted that "R_ex" represents the average scores predicted for the items. How does it manage to capture the variability in these scores? Have I missed something important?

This is a crucial logic in the paper that the authors should explain more clearly.

3. From an experimental perspective, why does the paper solely rely on the Gini index to measure unfairness? Isn't the regularizer R_ex used in the article to represent exposure inequality (although I disagree)?

4. It is advisable for this paper to explore the feasibility of discussing methods for other fairness definitions, such as user-side fairness, which may contribute more significantly to the field of fairness.

5. There is a lack of relevant baseline in the fairness domain. While Multi-FR and FairRec both emphasize multi-side fairness, this paper primarily focuses on item-side fairness. This work should involve more research and comparisons in the context of item-side fairness.

**Questions:**

See Weakness.

**Reviewer Confidence:**

4: The reviewer is certain that the evaluation is correct and very familiar with the relevant literature

**Scope:**

4: The work is relevant to the Web and to the track, and is of broad interest to the community

---

### Official Review · Reviewer_Gi34 · 2023-11-24

**Novelty:** 4
**Technical Quality:** 5

**Review:**

Pros:

1. The authors use theorems with detailed proofs to make the paper technically sound.

2. The authors provide an anonymous code repository that enhances reproducibility.

3. The authors emphasize their uniqueness by comparing it with previous works.

Cons:

1. It is not a good practice to list related works in detail in the Introduction Section. Probably demonstrate the meaning and importance of the research problems or topics by showing some cases.

2. It is worth considering whether this work is of broad interest to the community as it is an extension of the existing algorithm.

3. It would be better not to display the long equations in the text but separately, like the long equations in Theorem 3.1.

**Questions:**

1. Although this work is an extension of the celebrated iALS algorithm, can this method apply to other recommendation system models?

2. Based on Figure 1, the exADMM seems to underperform the "fair but inefficient" baselines. There should be some experiments to demonstrate the necessity of scalability. For example, the performance on a partial dataset is not enough to reflect that on the full dataset.

**Reviewer Confidence:**

2: The reviewer is willing to defend the evaluation, but it is likely that the reviewer did not understand parts of the paper

**Scope:**

3: The work is somewhat relevant to the Web and to the track, and is of narrow interest to a sub-community

---

### Official Review · Reviewer_iyEy · 2023-11-25

**Novelty:** 6
**Technical Quality:** 5

**Review:**

Strengths
- The paper proposes an evolution of iALS to account for exposure control aiming for a scalable formulation. I find it particularly positive that a focus is put on scalability which is often overlooked in academic research.
- Analysis is done on reasonably large datasets and the reported training time measurements support the claims made in the paper.

Weaknesses
- The experimental evaluation states that the train-validation-test split is done under a "strong generalization" setting which usually refers to user-holdout, which seems the case here. However, It is not explained how the interactions of the remaining 10% of users for validation-test are used. exADMM as iALS is a model-based recommender meaning that for cold users no user factors exist and one would need to either re-train the model or estimate the user factors based on the interactions of the user. This step is not explained. The authors should clarify how the split is performed.
- Some choices are made but not explained, for example different models are trained with a fixed number of epoch that is different (from 50 to 500) and the latent dimension used appears to be fixed for all methods, which is not a reliable way to conduct a comparison between different models and the embedding dimension is one of the most important hyperparameters.
- I appreciated that the comparison is done wrt strong baselines that do no optimize fairness and others that do but are not scalable. Unfortunately on MSD there seems to be no baseline optimizing fairness left, which makes the results on this dataset not very useful. Similarly on ML-20M there is essentially on 1 fairness baseline.


Given that Figure 1 is rather cluttered, I would suggest to remove the plots for MSD and display more clearly Epinions and ML-20M. Currently the y-axis is short and in limited space spans a large NDCG relative range.

**Questions:**

The authors should clarify how the data split is performed.

**Reviewer Confidence:**

2: The reviewer is willing to defend the evaluation, but it is likely that the reviewer did not understand parts of the paper

**Scope:**

3: The work is somewhat relevant to the Web and to the track, and is of narrow interest to a sub-community

---

### Official Review · Reviewer_5dvC · 2023-11-29

**Novelty:** 3
**Technical Quality:** 4

**Review:**

**Summary**
This paper proposes a computationally efficient method to improve the computational efficiency while guaranteeing exposure fairness so that the proposed method can be used flexibly on large-scale datasets. The real-world experiment verify both the effectiveness and efficiency of the proposed method.

**Pros**
- The research question is interesting and important.
- This paper is well-organized and clear written.
- This paper provides codes for reproducibility.
- The real-world experiments verify both the effectiveness and efficiency of the proposed method.

**Cons**
- Do the proposed method still valid when other popular fairness metrics are considered instead of constraining only the predicted exposure of item, e.g., differences in exposure between individual item $i$ and item $j$ ?
- Why is the NDCG@K obtained by the proposed method lower than Mult-VAE and iALS when the Gini@K are the same? The authors could consider other evaluation metric like Recall@K to further validate the effectiveness of the proposed method.

**Questions:**

Please refer to the **Cons** part for the questions.

**Reviewer Confidence:**

2: The reviewer is willing to defend the evaluation, but it is likely that the reviewer did not understand parts of the paper

**Scope:**

3: The work is somewhat relevant to the Web and to the track, and is of narrow interest to a sub-community

---

### Decision · Program_Chairs · 2024-01-22

**Decision:**

Accept

**Comment:**

Many reviewer concerns relate to experimental questions, which the authors have addressed on rebuttal. What I appreciate in the author-reviewer discussion is the carefulness with which the authors make their claims and acknowledge the advantages of other competing algorithms in some cases. Aside from the paper title itself, this indicates to me that the authors have done their best to provide fairly evaluated empirical results. One sticking point that remains, however, is raised by reviewer Rxja in their last comment that a more fair comparison of time would report time to convergence rather than the time for a fixed number of training epochs.

 One reviewer who has read the technical contributions closely finds that the writing and clarity of the presentation could be improved. While the authors have defended and clarified their claims, it would have been useful if the authors had stated how they would improve the presentation on revision to address reviewer concerns.

 > "the accuracy metric" refers to "NDCG@K,"

 For this comment by the authors, I note that accuracy has a well-defined meaning so redefining it as NDCG@K is confusing for the reader. I think the better response here would have been to accept the criticism and agree to clarify terminology.

 In conclusion, this paper appears to make novel technical and experimental contributions to improve scalability for large-scale recommender systems in the fairness optimization literature. Some minor concerns remain that could be fixed on revision if the authors are willing to consider those changes (which is not always clear from the rebuttal).